# Inducing, Detecting and Characterising Neural Modules: A Pipeline for Functional Interpretability in Reinforcement Learning

Anna Soligo [1]   Pietro Ferraro [1]   David Boyle [1]

## Abstract

Interpretability is crucial for ensuring RL systems align with human values. However, it remains challenging to achieve in complex decision making domains. Existing methods frequently attempt interpretability at the level of fundamental model units, such as neurons or decision nodes: an approach which scales poorly to large models. Here, we instead propose an approach to interpretability at the level of functional modularity. We show how encouraging sparsity and locality in network weights leads to the emergence of functional modules in RL policy networks. To detect these modules, we develop an extended Louvain algorithm which uses a novel 'correlation alignment' metric to overcome the limitations of standard network analysis techniques when applied to neural network architectures. Applying these methods to 2D and 3D MiniGrid environments reveals the consistent emergence of distinct navigational modules for different axes, and we further demonstrate how these functions can be validated through direct interventions on network weights prior to inference.

## 1. Introduction

Reinforcement learning (RL) has emerged as a powerful approach to improve performance in complex decision-making domains. Learning policies directly from interactions can offer improved flexibility and performance, whilst avoiding challenges faced by classical model based control approaches (Song et al., 2023). The growing body of RL research is demonstrating its potential to positively impact diverse real-world domains, from battery manufacturing (Lu et al., 2020) to the design of medical treatment regimes (Coronato et al., 2020), applications which directly impact critical issues such as climate-change and human-health.

However, this breadth of impacts also raises wide-ranging concerns related to topics of safety, reliability and bias, among others. It is thus crucial that the behaviour of RL agents can be properly characterised to the extent that it can be reasonably verified that their impacts align with human values. As reflected in the EU's AI ethics guidelines: systems should allow for human oversight, accountability and transparency (European Commission & High-Level Expert Group on AI, 2019).

Currently, there remain fundamental challenges to achieving this, and RL systems rarely afford sufficient interpretability. One factor is the ambiguity regarding what constitutes a suitable 'explanation' of a model. Lipton (2016) considers two parallel concepts: 'simulatability', the ease with which a human can predict a model's output from its input and explanation, and 'decomposability', the extent to which constituent components of a model are themselves interpretable. Doshi-Velez & Kim (2017) reiterate this with the concept of 'cognitive chunks', emphasising the need for model explanations to be tractable to human interpreters.

More concretely, interpretability can be considered in terms of the affordances it provides. In some cases, interpretability enables formal verification of safety-relevant capabilities (Bastani et al., 2018). However, in complex, incompletely-defined scenarios, it can instead offer insights which enable downstream safety-relevant tasks, such as system auditing or direct interventions to reduce undesirable behaviours (Kohler et al., 2024; Delfosse et al., 2024).

When scaling these affordances to complex domains, interpretability at the level of neurons, or other fundamental model units, becomes problematic: both due to their sheer quantity and because individual neurons are rarely semantically meaningful in isolation (Elhage et al., 2022). We address this challenge by taking a modular approach to interpretability. Modularity is fundamental in diverse biological architectures, including physical structures of the brain (Gazzaniga et al., 2018). Similarly, human-decision making can be considered as decomposable into modular processes (Eppe et al., 2022), suggesting that modularity may offer a natural framework for enhancing human understanding of complex systems.

---

[1]Imperial College London. Correspondence to: Anna Soligo <anna.soligo18@imperial.ac.uk>.

*Proceedings of the $42^{nd}$ International Conference on Machine Learning*, Vancouver, Canada. PMLR 267, 2025. Copyright 2025 by the author(s).

Motivated by this, we demonstrate how training modifications can encourage the emergence of functional modules within RL policy networks. We further propose methods to detect these modules and characterise their behaviour. In doing so, we aim to establish a suitable level of abstraction for aligning model interpretations with our internal decision making frameworks.

### 1.1. Contributions

Considering interpretability at the level of functional modules, our work makes the following contributions[1]:

- We extend recent algorithms for encouraging locality in neural networks (Margalit et al., 2024; Liu et al., 2023; Achterberg et al., 2023) to an RL context, demonstrating that penalizing non-local weights facilitates the emergence of functional modules within policy networks (Section 4.2). These modules offer a scalable unit for decomposing decision making, moving beyond interpretability at the level of neurons.

- We propose an extended Louvain algorithm for community detection which addresses the limitations of conventional community detection methods when applied to neural networks (Section 3.5). We thus demonstrate the ability to automatically identify functionally cohesive neural modules (Section 4.3) in a manner which enables the scaling of module based interpretability to complex networks.

- Utilising this approach to module detection, we demonstrate how targeted modifications of network parameters prior to inference can be used to characterise module behaviour, offering an empirical understanding of their functionality (Section 4.6).

## 2. Background

Following (Glanois et al., 2024), and to avoid confusion arising from the inconsistent use of terms in the literature, we denote interpretability as the extent to which a model's inner workings can be examined and understood. We distinguish this from explainability, which we define as an external understanding of model behaviour generally arising from post hoc attempts at explaining input-output relations. Interpretability and explainability present two approaches to obtaining information which can be used to form explanations for model-behaviour. Specifically, this work takes a direct interpretability approach, learning an inherently more interpretable model architecture.

Structural modularity, a property well-studied in network analysis, is characterised by the presence of communities

---

[1]All code is available at: https://github.com/annasoligo/BIXRL

of nodes with denser intra-community connections than inter-community ones. Detection of these communities is an NP-hard problem (Fortunato, 2010), and a multitude of methods have been developed to avoid the brute force approach, including hierarchical clustering, non-negative matrix factorisation (Lee & Seung, 2000), and the Louvain algorithm (Blondel et al., 2008). For the purpose of interpretability, we are further interested in functional modularity: the presence of components which show a level of independence and specialisation in their function (Fodor, 1985; Sternberg, 2011).

Functional modularity in the brain arises alongside its 'small-world' architecture: a combination of high clustering and short path length hypothesised to have evolved partially to satisfy spatial and energy constraints (Margalit et al., 2024). Recent works have investigated applying analogous constraints to neural networks. Liu et al. (2023); Achterberg et al. (2023); Margalit et al. (2024) demonstrate that penalising parameter 'connection length' in neural networks can lead to clustering and improved interpretability of network visualisations. We primarily build on the brain-inspired modular training approach proposed by Liu et al. (2023). We extend the concept of distance weighted regularisation to the RL context and further propose methods to extract and characterise functionally relevant modules from these regularised networks, enabling scalable interpretability in a decision making context.

## 3. Methods

### 3.1. Spatially Aware Regularisation

Regularisation approaches encourage sparsity by penalising the magnitude of model parameters. Following (Liu et al., 2023; Achterberg et al., 2023), we extend this to encourage local connectivity by projecting the neural network into Euclidian space and scaling weight penalties by the 'distance' between the neurons they connect.

For a network with $L$ weight layers, we denote neuron layers $\boldsymbol{N}_l$ for $l \in 0, \ldots L$, and weight matrices $\boldsymbol{W}_l$ for $l \in 0, \ldots L-1$. Each $\boldsymbol{W}_l \in \mathbb{R}^{n_l \times n_{l+1}}$ connects adjacent neuron layers, where $w_l^{ij}$ links the $i^{th}$ neuron in $\boldsymbol{N}_l$ to the $j^{th}$ neuron in $\boldsymbol{N}_{l+1}$. Each neuron, $n_l^i$, is assigned a 2D coordinate. To preserve their sequential nature, neurons within each layer $\boldsymbol{N}_l$ share a fixed y-coordinate $y_l^i := l$. Initial x-coordinates are uniformly spaced as $x_l^i = \frac{i}{n_l}$.

Standard L1 regularisation, $\sum_i^N |w_i|$, promotes sparsity by penalising the sum of absolute weight values. However, this scales linearly with weight magnitude, such that two weights of size $x$ incur the same penalty as a single weight of $2x$. We thus introduce a logarithmic sparsity loss, $\sum_i^N log(|w_i|+1)$ which provides a greater sparsity incentive by incurring a penalty which scales with $(x+1)^k$ rather than $kx$. We

provide further explanation and analysis in Appendix A. Scaling sparsity by distance gives the 'connection cost' loss:

$$L_{cc} = \lambda_{cc} \sum_{l=1}^{L} \sum_{i=1}^{n_{l-1}} \sum_{j=1}^{n_l} log((d_{ij} - d_s)|w_l^{ij}| + 1) \quad (1)$$

where

$$d_{i,j} = \sqrt{(x_j - x_i)^2 + (y_i + y_j)^2}$$

$\lambda_{cc}$ is the regularisation scaling factor, and $d_s$ adjusts the relative impacts of weight 'length' and magnitude.

### 3.2. Neuron Relocation

To further minimise $L_{cc}$, neurons are periodically relocated during training, following (Liu et al., 2023). Within each layer, neurons are ranked by their weighted degree $w(n) = \sum |w_{in}| + \sum |w_{out}|$. The top $k$ neurons are optimised within their layer by exchanging their position with the alternative neuron position which leads to the greatest reduction in $L_{cc}$, as detailed in Algorithm 1. This has the effect of changing the relative 'cost' of weights, such that weights with a greater impact on performance can be retained with a lower relative connection cost. We discuss this further in Appendix E.2.

---

**Algorithm 1** Neuron Position Optimization

---

1: **Every** $T_{swap}$ **training steps:**
2: **for** each layer $l$ in $[1, L]$ **do**
3:     Calculate weighted degrees: $w(n) = \sum |w_{in}| + \sum |w_{out}|$ for all $n \in \mathbf{N}_l$
4:     Select top $k$ neurons with highest $w(n)$
5:     **for** each candidate neuron $n_l^c$ **do**
6:         Compute baseline cost $L_{cc}^0$ using Equation 1
7:         $L_{cc}^{best} \leftarrow L_{cc}^0$,   $n_l^{best} \leftarrow$ None
8:         **for** each neuron $n_l^i$ in layer $l$ **do**
9:             Calculate $L_{cc}^i$ after swapping positions of $n_l^c$ and $n_l^i$
10:             **if** $L_{cc}^i < L_{cc}^{best}$ **then**
11:                 $L_{cc}^{best} \leftarrow L_{cc}^i$,   $n_l^{best} \leftarrow n_l^i$
12:             **end if**
13:         **end for**
14:         **if** $n_l^{best}$ is not None **then**
15:             Swap positions of $n_l^c$ and $n_l^{best}$
16:         **end if**
17:     **end for**
18: **end for**

---

### 3.3. Structural Modularity in Networks

In network analysis, structural modularity is quantified by comparing the strength of intra-community links with their expected strength in a random 'null model', such that high modularity indicates stronger connectivity within a set of defined modules than would occur by chance (Clauset et al., 2004). Given the network partition $P = \{C_1, C_2, \ldots, C_k\}$

where the community of node $n^i$ is denoted $C(n^i) \in P$, modularity $Q$ is defined as:

$$Q = \frac{1}{2m} \sum_{ij \in P} \left( A_{ij} - \frac{k_i k_j}{2m} \right) \delta(c_i, c_j) \quad (2)$$

where $m = \sum_{lij} w_l^{ij}$ is the sum of all edge weights, $A$ is the network adjacency matrix, $k_i$ is the weighted degree of node $i$ and the binary $\delta(c_i, c_j)$ is a binary variable which equals 1 if nodes $i, j$ share a community. The null model $\frac{k_i k_j}{2m}$ models random connectivity given node orders and acts as the non-modular baseline. This equation forms the basis of the heuristic Louvain algorithm (Blondel et al., 2008), which optimises $P$ to maximise $Q$ through hierarchical local node reassignments.

We take the Louvain algorithm as a baseline partitioning approach due to its efficiency ($O(nlog(n))$), automatic detection of community number, and relative simplicity. However application to neural networks reveals limitations arising from the differences between NNs and traditional networks. Primarily, the high fan-out connectivity of input features and constrained layer-wise connectivity in MLPs results in Louvain partitions which fail to span sufficient weight layers and violate the directionality of NN information processing. We further discuss and provide of this in Appendix B.

### 3.4. Modularity in Feed-forward Neural Networks

To address these challenges, we propose two metrics to quantify neural network modularity while accounting for their architecture and the specific utility of modularity for interpretability. Firstly, we consider module isolation. High isolation implies minimal inter-module connectivity, resulting in stricter decomposability and enabling more independent module analysis. For a single module, we define isolation $I(C)$ as:

$$\mathrm{I}(C) = \frac{\mathbf{W}_{int}}{\mathbf{W}_{int} + \mathbf{W}_{ext}} \quad (3)$$

where $\mathbf{W}_{int} = \sum_{i,j \in C} |w^{ij}|$ $\mathbf{W}_{ext} = \sum_{i \in C, j \notin C} |w^{ij}| + \sum_{i \notin C, j \in C} |w^{ij}|$ represents the sum of intra- and inter-community weights respectively. We extend this to the isolation of a network partition $P$:

$$\mathrm{I}(P) = \begin{cases} 0 & \text{if } |P| = 1 \\ \frac{1}{|P|} \sum_{C \in P} \mathrm{I}(C) & \text{otherwise} \end{cases} \quad (4)$$

Secondly, we consider the alignment between structural and functional modularity by considering correlations between neuron activations. Neuronal correlations have been used to study functional architectures of biological neural networks (Cohen & Kohn, 2011), as well as similarities between artificial neurons (Li et al., 2016). We calculate the Pearson

correlation coefficients $r^{ij}$ between each pair of neurons $i, j$, based on their activations $n^i(t)$ and $n^j(t)$ over $T$ samples.

$$r^{ij} = \frac{\sum_{t=1}^{T}(n^i(t) - \bar{n}^i)(n^j(t) - \bar{n}^j)}{\sqrt{\sum_{t=1}^{T}(n^i(t) - \bar{n}^i)^2}\sqrt{\sum_{t=1}^{T}(n^j(t) - \bar{n}^j)^2}} \quad (5)$$

These correlation values form the adjacency matrix of a functional network graph, $G_F$, in which, unlike the weight-based structural network graph, $G_S$, connections are not constrained to adjacent layers. Given two Louvain partitions, $P_F = \{F_1, F_2, ..., F_m\}$ and $P_S = \{S_1, S_2, ..., S_n\}$, of these graphs, the Adjusted Rand Index (ARI) quantifies their similarity and thus the 'correlation alignment' of $P_S$:

$$ARI(P_F, P_S) = \frac{2\sum_{ij}\binom{n_{ij}}{2} - \left[\sum_i \binom{s_i}{2} \sum_j \binom{f_j}{2}\right]}{\sum_i \binom{s_i}{2} + \sum_j \binom{f_j}{2} - 2\sum_{ij}\binom{n_{ij}}{2}} \quad (6)$$

where $n_{ij}$ is the number of nodes shared between modules $i$ of $P_F$ and $j$ of $P_S$, and $f_i$ and $s_j$ are the total numbers of nodes in modules $i$ and $j$ respectively.

### 3.5. Detecting Modules in Neural Networks

Utilising these isolation and correlation alignment metrics, we propose a 'fine-tuning' stage which improves the initial Louvain partition $P_S$ by iteratively merging modules to maximise the structural and functional modularity of the resulting partition. As detailed in Algorithm 2, functional and structural partitions, $P_F$ and $P_S$, are initialised using an 'internal' variation of the Louvain algorithm. This excludes input layer nodes in the initial partitioning, then subsequently assigns them to the community to which they are most strongly connected, mitigating the challenges the input layer poses to module detection. $P_S$ is evaluated according to its modularity score $M = I(P_S) + ARI(P_S, P_F)$, and adjacent modules are merged in the manner that maximises $M$, until no further improvement is obtained.

## 4. Experiments

We evaluate the proposed methods with respect to 4 main research questions. These examine the ability to induce (RQ1), detect (RQ2) and interpret (RQ4) modularity, while considering auxiliary impact on model performance (RQ3).

**RQ1.** Does spatially aware regularization lead to the emergence of modular structures in RL policy networks?
**RQ2.** Can emergent modular structures be detected using the proposed extended Louvain algorithm?
**RQ3.** What is the quantitative relationship between RL policy modularity and performance?
**RQ4.** Do detected network modules correspond to

---

**Algorithm 2** Interpretability Fine-tuning of MLP Modules

1: Initialize $P_{S,current}$ and $P_F$ using the Louvain algorithm
2: In each of $P_{S,current}$ and $P_F$, assign each input neuron $n_0^i$ to the community to which it is most strongly connected.
3: Calculate initial modularity score $M = I(P_{S,current}) + ARI(P_{S,current}, P_F)$
4: **repeat**
5:    **for** each pair $(P_i, P_j)$ of adjacent modules **do**
6:       Calculate $M_{ij}$ for merged modules
7:    **end for**
8:    **if** $\max(M_{ij}) > M$ **then**
9:       Update $P_{S,current}$ with best merge
10:       $M \leftarrow \max(M_{ij})$
11:    **end if**
12: **until** $\max(M_{ij}) < M$

Figure 1: The Dynamic Obstacles (DO) and Go to Key (G2K) Environments.

interpretable and functionally relevant components?

### 4.1. Experimental Setup

Experiments are conducted in three Minigrid environments (Chevalier-Boisvert et al., 2023; Pignatelli et al., 2024), shown in Figure 1: Go-to-key (G2K), where an agent must navigate to one of two keys in a 4x4 grid; dynamic obstacles (DO), where an agent must reach a goal in a 4x4 grid whilst avoiding three moving obstacles; and 3D dynamic obstacles (3D-DO), which extends dynamic obstacles to a 3x3x2 grid. These are encoded into a symbolic observation of entity coordinates relative to the agent. The action space consists of left, right, up, down, and, in the 3D case, forward, backward steps. Following Pignatelli et al. (2024), a Markov reward function offers a sparse reward of 1 when the target is reached and 0 otherwise. The reported returns thus represent both the mean episode return and the success rate. Episodes terminate on goal completion, collision with obstacles, reaching the incorrect key, or exceeding the 100 step limit.

We train all policies using Proximal Policy Optimisation (Schulman et al., 2017) due to its stability and simplicity. The actor and critic networks are implemented as MLPs with two hidden layers of 32 neurons and hyperparameters (Table 1) optimised via grid search. We focus on decision making interpretability, and apply distance weighted regularisation to the actor network. The regularisation coefficient

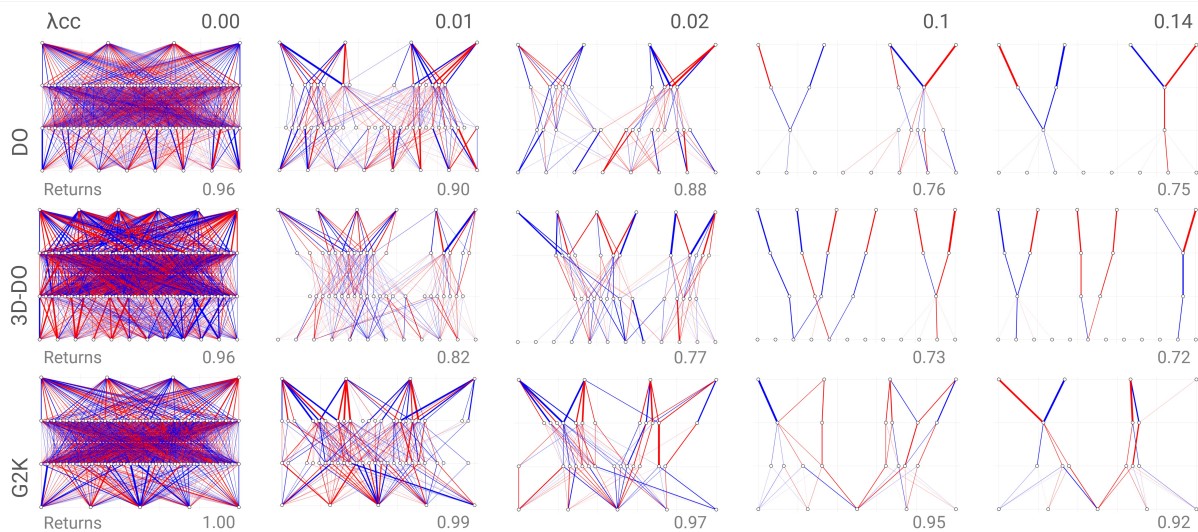

Figure 2: Distance weighted regularisation induces the emergence of visual modularity. As $\lambda_{cc}$ is increased, increasingly isolated modular structures are observed in the policy networks of the DO (top), 3DO (middle) and G2K (bottom) environments. A moderate decrease in mean return is also observed, as annotated below each network plot.

$\lambda_{cc}$ is increased linearly from 0 to its target value between 20 and 30% of training steps. Agents are trained for 4M environment frames, pruned by removing weights and neurons with magnitudes below 1% of the maximum values and orders in their respective layers, then fine-tuned without $L_{cc}$ regularization for 2M frames. This regularization schedule and two-stage training methodology yields improved returns and modularity metrics, as detailed in Appendix C. The PPO agent and environment are implemented in JAX (Bradbury et al., 2018) and trained using a NVIDIA RTX 4090 GPU.

### 4.2. Emergence of Visual Modularity (RQ1)

Structural modularity emerges as the strength of distance weighted regularisation is increased. As shown in Figure 2, module independence initially emerges in the second and third weight layers, while feature sharing persists in the first. Across all environments, neuron relocation causes the input features and output actions to reorder in a manner that reflects their relevance. Figure 3 shows how feature x, y and z coordinates align vertically with the actions controlling movements on the x, y and z axes respectively. In the DO and 3D-DO environments, modules become fully independent at high $\lambda_{cc}$ values, and networks increasingly prioritise goal features over obstacles. Conversely, the target key ID in the Go to Key task remains used by both navigational modules at all $\lambda_{cc}$ values, reflecting its necessity to solve the task. Figure 4 shows the importance of both the connection cost loss and neuron relocation in inducing this modularity and we provide further ablation results isolating their impacts in Appendix E.

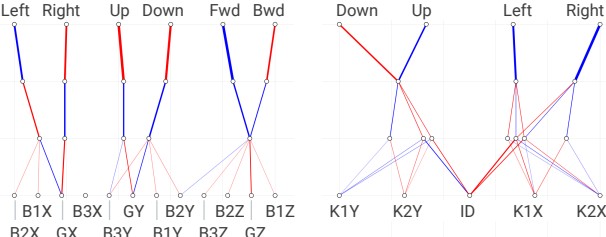

Figure 3: Neuron relocation causes input features and output actions align spatially by function. In the 3D-DO (left) and G2K (right) policy networks, object coordinates align spatially with the actions controlling movements along their corresponding axes.

### 4.3. Module Detection (RQ2)

We benchmark our proposed fine-tuned internal-Louvain approach against the standard Louvain algorithm, and further evaluate the isolated impacts of the fine-tuning and internal-Louvain modifications. Fine-tuning (FT) significantly increases the average isolation (16.8%) and correlation alignment (33.7%) of the detected modules across all $\lambda_{cc}$ values (as detailed in Appendix G). Initialising with the internal Louvain (FT Int.) increases isolation by a further 2.4% but decreases correlation alignment by 1.6%. Notably, this disparity in ARI between the FT and FT Int. methods predominantly arises from differences in the G2K results. While the FT method gives partition ARIs that reach a maximum value at $\lambda_{cc} = 0.11$ before decreasing, the FT Int. method gives a monotonically increasing ARI, which exceeds the FT ARI for $\lambda_{cc} > 0.07$. Figures 5 and 6 exemplify these improvements in performance at varying $\lambda_{cc}$ values.

The G2K networks retain feature sharing between modules

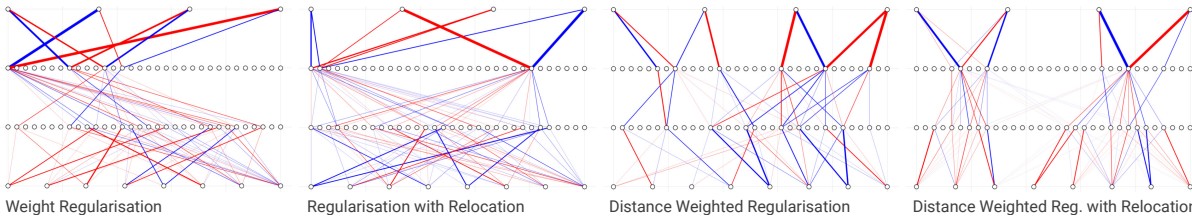

Weight Regularisation          Regularisation with Relocation          Distance Weighted Regularisation          Distance Weighted Reg. with Relocation

Figure 4: The combination of distance weighted regularisation and neuron relocation results in the most modular networks. Ablating the distance weighting (left and second left) or the relocation (left and second right) does not achieve the same level of modularity. We provide further quantitative ablation results in Appendix E.2.

in highly regularised networks. This, as detailed in Section 3.3 and Appendix B, presents a failure case for the standard Louvain algorithm, whereby it fails to distinguish modules within the first layer, as can be seen in Figure 6. Since the correlation partitions are less clearly segregated in networks with higher connectivity, and occasionally exhibit this input layer failure, we deem ARI to be a less reliable indicator of modularity at lower $\lambda_{cc}$ values, and therefore adopt the fine-tuned internal method.

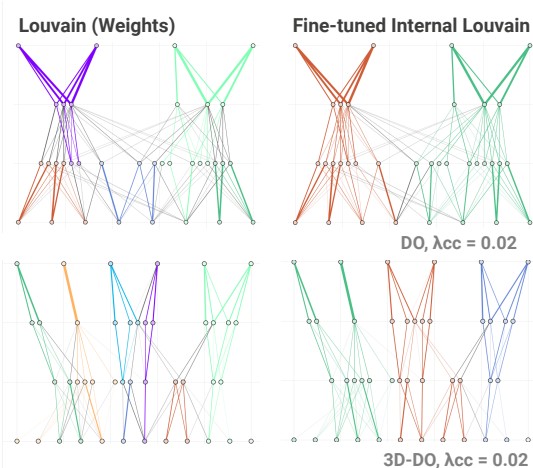

Figure 5: The modified Louvain algorithm is able to identify functional modules across multiple layers. Our fine-tuned internal Louvain method (right) successfully detects cohesive neural modules, whereas the standard Louvain (left) incorrectly subdivides modules in DO (top) and 3D-DO (bottom) policy networks due to limitations in handling MLP architectures.

## 4.4. Quantification of Modularity (RQ1, RQ3)

Applying the FT. Int. partitioning method, we find module isolation and correlation alignment increase with $\lambda_{cc}$, as shown in Figure 7. The induced sparsity, does, however, impact negatively on return. As shown in Figure 8, we find this impact varies significantly by environment. At the point of module emergence, returns decrease by an average of 11.4% and 12.5% for DO and 3D-DO respectively ($\lambda_{cc} \approx 0.02, 0.04$), compared to just 0.8% for G2K ($\lambda_{cc} \approx 0.02$). We also find that implementing regularisation and neuron relocation results in an average training time increase of

17%, which, as detailed in Appendix D.1, is largely due to the regularisation component.

Despite this performance trade-off, we note that our regularisation approach offers an auxiliary benefit by yielding much smaller networks. At the 'modularity emergence' stage, our final DO, 3D-DO and G2K networks have 90.5%, 96.5% and 89% fewer parameters, respectively, than those trained without regularisation. This offers a means of significantly reducing computational overhead during inference in addition to further potential interpretability benefits.

## 4.5. Functional Interpretability (RQ4)

While visualisation of network modules offers a level of insight into the structure of decision making, it relies on subjective assessment and lacks scalability. Consequently, we use targeted modification of parameters as an empirical means of interpreting module functionality.

For the example networks shown in Figure 9, we systematically modify module parameters in two ways: negative saturation, in which we replace all values with a large negative value of -50; and negation, where we reverse the sign on all parameters. The former aims to effectively disable a module, while the latter aims to perturb it. We evaluate the subsequent behavioural changes over 10,000 episodes, measuring the frequency of actions and their corresponding outcomes: success, failure, or continuation of the episode.

Foremost, we find that negatively saturating any individual module strongly inhibits actions along a specific axis. This validates that the detected structural modules correspond to axis specific navigation. Intervening on community 0 in Figure 9a, for example, reduces the frequency of forward/backward actions by 83%, while intervening on community 0 in Figure 9b reduces the frequency of up/down actions by 95%. These results replicate, with slightly reduced functional independence, in less regularised networks like Figure 9c, where community 0 intervention results in a 91% decrease in up/down actions, while community 1 intervention results in a lesser 42% decrease in left/right. Notably, while the overall success rate decreases when we saturate modules in the dynamic obstacles environments,

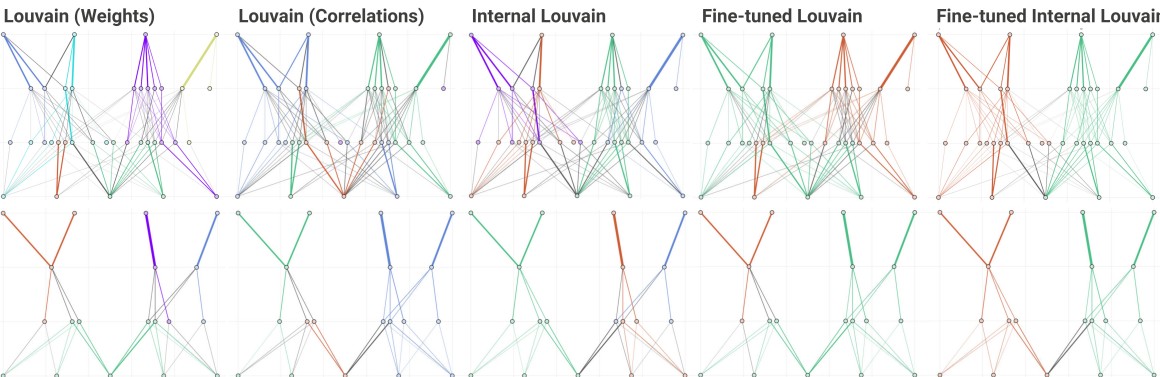

Figure 6: The internal Louvain and fine-tuning stage result in modules which are more isolated and better aligned with the activation-based network partition. Utilising the internal Louvain improves module assignment in the input layer, particular in the lesser-regularised example (top), while the fine-tuning stage reduces the subdivision of modules between layers.

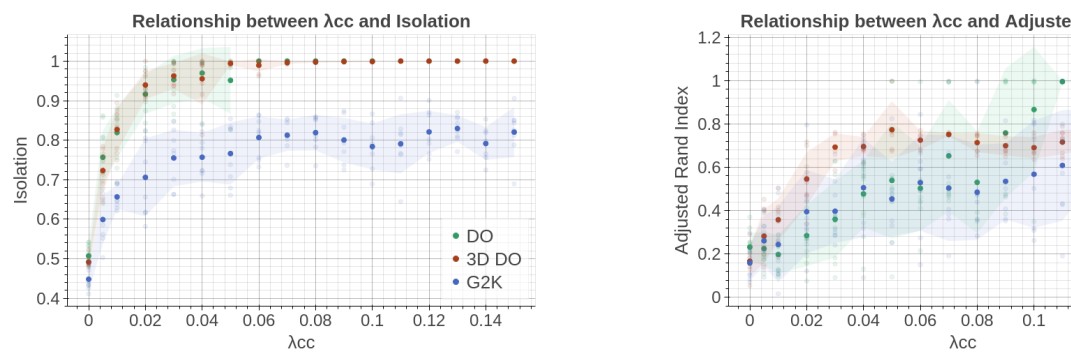

Figure 7: Increasing regularisation strength results in increased module isolation and correlation alignment (ARI). The mean and standard deviation (n = 10) of isolation (left) and ARI (right) increase with $\lambda_{cc}$.

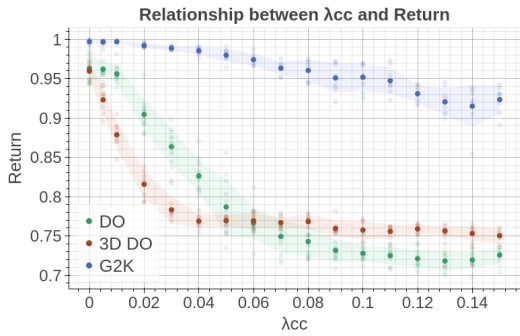

Figure 8: The performance trade-offs observed with increased regularisation strengths vary across environments. The mean and standard deviation of return (n=10) shows varying levels of performance degradation as $\lambda_{cc}$ is increased in the three environments.

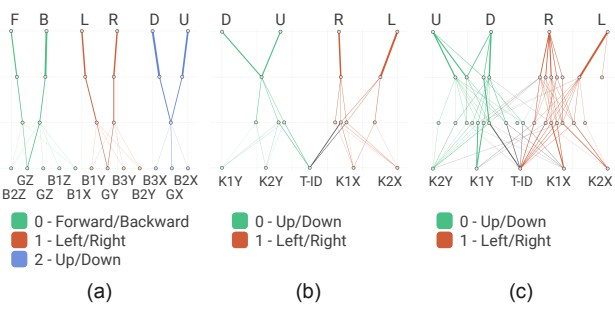

Figure 9: The detected modules specialise in navigation along a specific axes. Examples of partitioned policy networks for (a) 3DDO ($\lambda_{cc}$ = 0.06), (b) G2K ($\lambda_{cc}$ = 0.12), (b) G2K ($\lambda_{cc}$ = 0.02), with the identified module functions described in the legends.

the proportion of actions resulting in failures does not increase. This shows that with the achieved level of functional independence, we can disable a module while retaining the decision-making ability of those remaining.

Compared to negative saturation, negation has a significantly stronger impact on return: rather than minimising actions along a given axes, the agent now acts incorrectly. For the 3D-DO network presented, we observe an average decrease

in return of 73% when a module is negated compared to a decrease of only 36% when a module is negatively saturated. This also reflects in the failure rate: when community 0 in Figure 9b is negatively saturated, we observe that the ratio of success to failure outcomes of up/down actions declines from 37:1 to 9:1. When it is instead negated, this drops to 1:103. Full intervention statistics are given in Appendix H.

### 4.6. Learning Robust Pong Polices

We additionally train a Pong policy using the same PPO training protocol as the MiniGrid experiments. Due to the simplicity of navigation in Pong, this learns a single sparse module rather than multiple modules as we observe in the Dynamic Obstacle and Go to Key tasks. However, we find that that distance weighted regularisation improves visual interpretability of the network, and enables identification of a flaw in the learnt policy. We find that the sparse Pong policy network retains strong connectivity to the opponents position, which is a consequence of the opponent's 'follow ball' policy and was previously observed by (Delfosse et al., 2024). This reliance means the agent is not robust to changes in opponent policy. We consequently retrain robust Pong policies by removing the opponent position from the observation space, and note only minor decreases in performance. We fully detail these results in Appendix F.

## 5. Related Work

Existing direct interpretability approaches frequently rely on making fundamental architectural changes in order to build policies from intrinsically interpretable units. Examples include representing policies with differentiable 'soft decision trees', (Silva et al., 2020), symbolic equations (Hein et al., 2017; Landajuela et al., 2021), or weighted combinations of logic rules (Jiang & Luo, 2019; Delfosse et al., 2023). Recent works have extended these frameworks to remove previous barriers to their adoption in RL, for example by enabling on-policy learning of decision trees (Marton et al., 2025), by combining interpretable policies with deep-neural policies to improve performance (Shindo et al., 2024), and by harnessing large language models to improve downstream human interpretability (Luo et al., 2024). However, these methods continue to face scalability challenges, becoming computationally prohibitive in complex scenarios even when indirect policy distillation frameworks are adopted (Glanois et al., 2024). Moreover, interpretability at the level of fundamental model units is rapidly compromised by scaling: a decision tree with an intractable number of nodes, for instance, may be no more interpretable than a network with an intractable number of neurons.

Tangentially, the field of mechanistic interpretability takes a bottom up approach to reverse-engineering neural networks, particularly large language models. This can involve the identification of features (Templeton et al., 2024), concept representations (Zou et al., 2023) or computational 'circuits' (Wang et al., 2022). We share the behavioural focus of circuits work, but rather than attempting interpretability at the level of computations, we aim to characterise the roles of higher-level communities of neurons. Notably, recent work has partially automated the circuit-discovery process in transformers (Conmy et al., 2023). This, like our automated module detection, is motivated by the need to improve the scalability of interpretability techniques.

Modularity in RL is approached by hierarchical RL, particularly policy tree methods where decision making is decomposed into sub-policies (Pateria et al., 2021). These rely on predefined levels of decomposition, but can afford interpretability when discernible sub-behaviours, such as motor primitives (Merel et al., 2018), are explicitly learnt. More intentionally, Cloud et al. (2024) recently proposed to localise network computations through selectively masking parameter gradients. In contrast to our approach, this gradient-routing approach requires user-defined sets of parameters and data points to control the functional localisation process.

Prior work has explored applying biologically inspired connection constraints to neural networks. Achterberg et al. (2023) spatially embed an RNN and penalise connection length, demonstrating energy efficiency and clustering in a one-step inference task. Concurrently, Margalit et al. (2024) proposed to encourage local activation correlation in the training of network layers projected onto simulated cortical sheets. While these studies target the advancement of neuro-scientific understanding, Liu et al. (2023) aim to improve the interpretability of network visualisations. They demonstrate that length-relative weight penalisation reveals structure within regression and classification tasks such as learning mathematical formulae.

Community detection in graphs is a significant area of research with relevance to multiple disciplines, including computer science and biology (Fortunato, 2010). Although recent works adapt classical clustering approaches to specialised structures such as multiplex networks (Huang et al., 2021), the challenge of detecting communities within neural networks remains largely unexplored. Filan et al. (2021), study the extent to which non-regularised MLPs can be clustered, and Hod et al. (2022) extend this to determine whether such clusters are more 'coherent' than random sets of neurons. Both rely on spectral clustering (Shi & Malik, 2000), however, which is impractical for large neural networks due to its reliance on computing eigenvectors and predefining the number of communities.

# 6. Discussion

**Interpretability.** Our work addresses interpretability at the level of functional modules, targeting a level of abstraction that may offer a suitable balance of tractability and fidelity when scaled to large models. Given a model with emergent modules, we demonstrate how modular functionality can be systematically characterised through targeted weight interventions, enabled by our neural-network specific partitioning approach. We successfully identify module functions, but emphasize that this is a preliminary demonstration of module characterisation in simple environments. Other methods may offer improved insights, notably through activation rather than parameter modifications. This will become particularly relevant in complex architectures and applications, where we expect to achieve less module independence, and where preserving module output distributions will be necessary to preserve the downstream functionality.

As introduced in Section 1, interpretability can, in different contexts, enable both formal verification and safety auditing. While we primarily target the latter, due to its broader applicability at scale and given incomplete problem definitions, our sparse, modular approach could also advance formal verification. The extraction of relatively independent modules enables verification to be performed in isolation, reducing complexity and simplifying identification of failures.

**Scalability.** Scalability poses a fundamental challenge to interpretability, and we have aimed to address this at multiple levels. By preserving standard neural network architectures and training, we avoid the scaling limitations faced by white-box model approaches such as decision-tree or logic-based policies. By automating the classification of neurons into modules, we remove reliance on manual approaches to module detection. The subsequent characterisation of module functions through parameter modifications further automates the interpretability process, enabling scalability to complex models embedded in a higher dimensions than the two we consider in this work. Finally, by targeting decomposability at the level of functional modules rather than fundamental units such as neurons, we aim to maintain tractability as model complexity increases, thereby offering a level of simulatability that scales with network size.

**Limitations and Future Work.** The observed trade-off between interpretability and performance, while common among 'white-box' approaches, is undesirable and a barrier to the adoption of interpretable systems. The $\lambda_{cc}$ scaling factor offers a means of tailoring the level of interpretability based on specific requirements, but further investigation into mitigating the performance decline is warranted.

We offer a proof-of-concept in three environments, but demonstration in complex domains and agent architectures remains a key direction for future work. While we focus on the RL context, motivated by the relevance of modularity in decision making, this modular approach could be applied more broadly. We also note that while we induce and detect a high level of modularity in our examples, a lower level of spatially aware regularisation may be useful to promote sparsity and functional localisation in a manner which would improve the performance of post-hoc interpretability and explainability approaches.

Currently, the lack of formal metrics for interpretability makes it challenging to comparatively evaluate the utility of different interpretability methods. While certain metrics, such as performance, can be objectively measured, the critical notions of interpretation accuracy and tractability still lack rigorous means of evaluation, and the formalisation of these metrics poses an important challenge for advancing interpretable AI. In their absence, we discuss performance, tractability and scope of our approach. Future work exploring specific, real-world use cases could examine how verification or user-studies could be used to validate the utility of this interpretability approach.

# 7. Conclusion

In this work, we have demonstrated how spatially aware regularisation induces the emergence of structural and functional modularity in the policy networks of RL agents. We develop a novel approach and metrics to quantify and detect modularity in neural networks, and leveraging this, automatically identify and characterise decision-making modules. By addressing interpretability at the level of functional modules rather than fundamental units, we offer a promising balance between fidelity and human tractability. Future work should explore the broad potential applicability of this approach within different model architectures and its scalability to complex, real-world domains.

## Acknowledgments

The authors thank the IOTA Foundation, Google and UKRI EPSRC [grant numbers EP/Y037421/1 and EP/X040518/1] for supporting this research.

## Impact Statement

This work advances the interpretability of deep reinforcement learning systems, with direct implications for the safe deployment of RL in real-world applications. The proposed methods could contribute to enhancing human oversight and verification of AI systems through increasing understanding of decision making processes. In addition to enhancing safety, this aligns with growing regulatory requirements and could help accelerate the adoption of RL in safety-critical domains.

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

# A. Sparsity Methods

The L1 norm ($\sum_i^N x_i$) is known to induce sparser solutions than the L2 norm ($\sqrt{(\sum_i^N x_i^2)}$) due to its having constant gradients with respect to parameter magnitudes. In contrast the gradient of the L2 norm decreases with its magnitude, resulting in an optimisation landscape that preferentially reduces larger parameters rather than promoting sparsity through the elimination of near-zero parameters. However, while an L1 loss function does not directly discourage reducing near-zero parameters, nor does it favour it: the L1 norm optimises for minimal total parameter magnitude, rather than a low count of non-zero parameters, which is what we desire in a sparse model.

In contrast, our proposed log-based sparsity loss, $\sum_i^N log(|x_i| + 1)$, has a gradient $\frac{\partial}{\partial x_i} = \frac{1}{x_i+1}$. This scales inversely with parameter magnitude, thus explicitly promoting sparsity. We note an alternative formulation $exp(\sum_i^N log(|x_i| + 1))$, with gradient $\frac{\partial}{\partial x_i} = \prod_{j \neq i}^N x_j$, which similarly directly encourages sparsity as a result of having relatively higher gradients for parameters whose magnitudes are low in the parameter distribution. We adopt the former due to its linear scaling with respect to number of parameters.

The comparative behaviour of these functions can also be intuitively understood by observing the gradients and value of the contour plots in Figure 10 when varying $x$ and $y$.

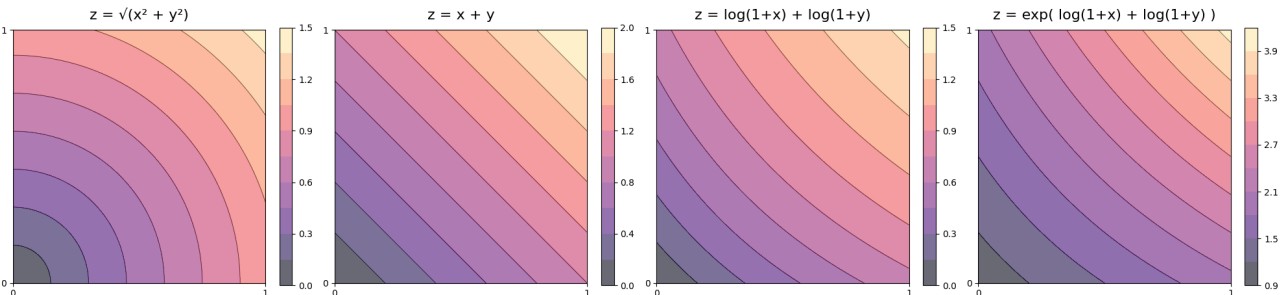

Figure 10: Contour plots showing the results of L2, L1, log and exponential-log loss functions for two parameters.

In our work, we find that utilising the log rather than L1 based regularisation results in a preferable relationship between return and isolation, and between return and ARI, as shown in Figure 11.

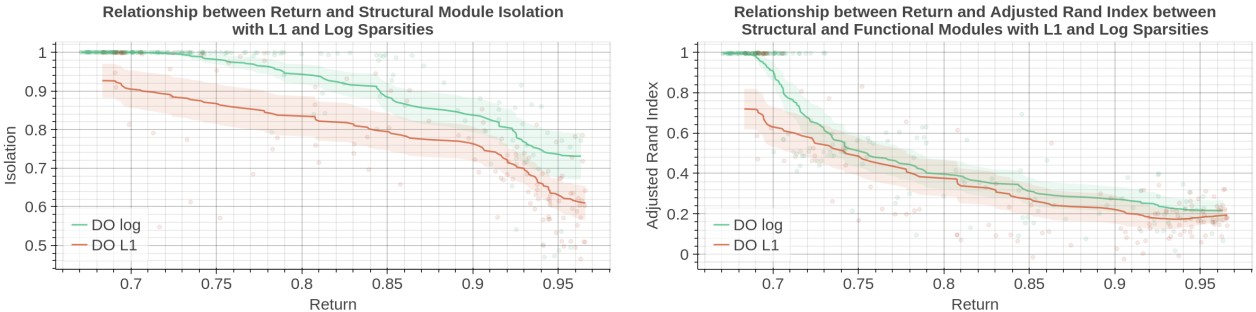

Figure 11: The relationship between return and module isolation (left) and between return and ARI (right) for models trained with L1 and log based regularisation, showing that log based sparsity results in more isolated and functionally aligned modules.

## B. Applying the Standard Louvain Algorithm to MLP networks

The constrained layer-wise connectivity of MLPs conflicts with the null-model assumption of a uniform connection probability between nodes. The Louvain equation (Equation 2) compares the magnitude of a network connection with its expected strength within a random network with the same node orders ($\frac{k_i k_j}{2m}$). Unlike the arbitrary sub-graphs observed in traditionally modular networks, neural modules form continuously across multiple adjacent layers, with connections only present between consecutive layers. They thus generally exhibit a lower connection density, leading to modules becoming subdivided. Additionally, neural network input layers frequently show high fan-out connectivity patterns where feature information is distributed to multiple downstream neurons. The resulting areas of high connectivity satisfy $Q$ optimization, despite spanning a single weight layer and violating the directionality of information processing in the network.

We illustrate these issues in Figure 12, which contrasts a classically modular network architecture with examples of modular MLP networks. While the Louvain algorithm performs well for the former, several 'failure cases' arise in the latter. The green modules exemplify the challenge of distinguishing modules when feature sharing and fan out connectivity occur in the input layer. Furthermore, few of the Louvain detected modules span the full depth of the network, despite it being visually apparent that the network modules do so, which results in the network modules being subdivided.

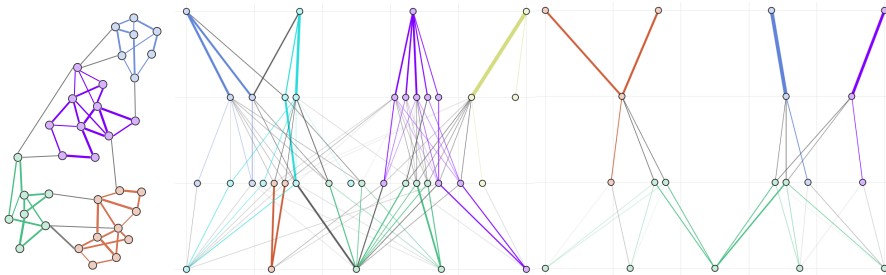

Figure 12: Examples of the clustering results of the Louvain algorithm when applied to a modular network without layer-wise structural constraints (left), and to modular MLPs. Note the modules (in green) which span multiple modules in the input layer, and the modules which fail to span the full module depth.

# C. Hyperparameter and Training Choices

Table 1: PPO Hyperparameters

| ARCHITECTURE | |
|---|---|
| HIDDEN SIZE | 32 |
| NUMBER OF LAYERS | 2 |

| TRAINING | |
|---|---|
| PARALLEL ENVIRONMENTS | 16 |
| STEPS PER ENVIRONMENT | 128 |
| MINIBATCHES | 8 |
| EPOCHS | 16 |
| LEARNING RATE | 5E-4 |
| MAX GRADIENT NORM | 0.5 |
| GAE $\lambda$ | 0.99 |
| CLIP $\epsilon$ | 0.2 |
| ENTROPY COEFFICIENT | 0.01 |
| VALUE FUNCTION COEFFICIENT | 0.5 |

| REGULARISATION | |
|---|---|
| $d_s$ (EQUATION 1) | 0.95 |
| $\kappa$ (SECTION 3.2) | 10 |
| RELOCATION INTERVAL (SECTION 3.2) | 2 |

## C.1. Pruning and Fine-tuning

We prune our networks after 80% of the training steps, and train for the remaining 20% with $\lambda_{cc} = 0$. Pruning involves setting all weights with a value below 1% of the maximum absolute weight in their layer to 0, and removing all hidden-layer neurons with an order (total sum of incoming and outgoing weights) of less than 1% of the maximum in their layer. Gradients for the removed parameters are masked at 0 for the remainder of training. The pruning fixes a sparse and modular architecture, and we find the fine-tuning with no regularisation improves performance (Figure 13), does not compromise the isolation the modules (Figure 15), and slightly increase their ARI (Figure 16). We select a 1% pruning level because this does not result in a significantly reduced return compared to lower pruning levels, offers a higher level of resulting sparsity (Figure 14), and results in the modules with the highest ARI.

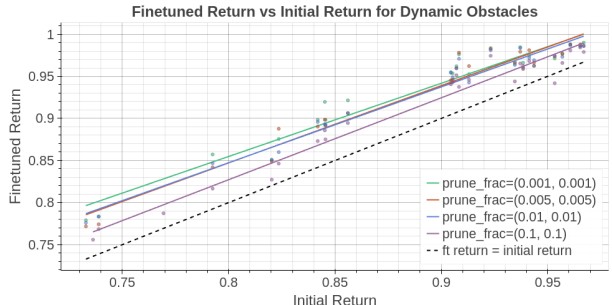

Figure 13: The relationship between initial return (before pruning and fine-tuning) and post fine-tuning return for different pruning thresholds applied to DO networks with $\lambda_{cc} \in [0.005, 0.1]$.

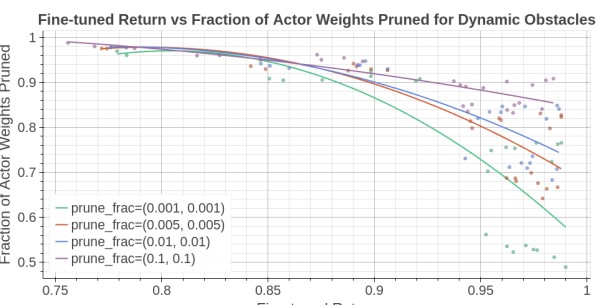

Figure 14: The fraction of actor weights pruned and the fine-tuned return achieved with different pruning levels applied to networks with $\lambda_{cc} \in [0.005, 0.1]$.

## C.2. Selecting $d_s$

As introduced in Section 3.1, the value of $d_s$ varies the relative significance of distance and weight in regularisation. Primarily, given the range of possible distances between neurons in adjacent layers $d_s \in [1, \sqrt{2}]$, a value of $d_s = 1$ means that a weight connecting two vertically aligned neurons contributes 0 to the total connection cost. This is apparent in Figure 17, where networks with $d_s = 1$ have a high number of these 'vertical' weights. In contrast, as $d_s$ is decreased, we see an increasing number of distant connections in the network.

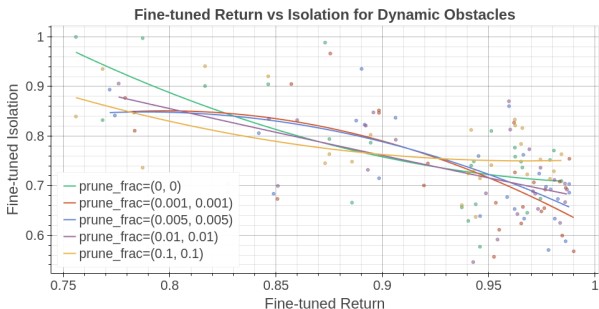

Figure 15: The relationship between fine-tuned return and isolation for different pruning levels. With a pruning level of 0, we do not set $\lambda_{cc} = 0$ for fine-tuning, as this allow all weights in the fully connected network to increase, compromising modularity.

Figure 16: The relationship between fine-tuned return and ARI for different pruning levels.

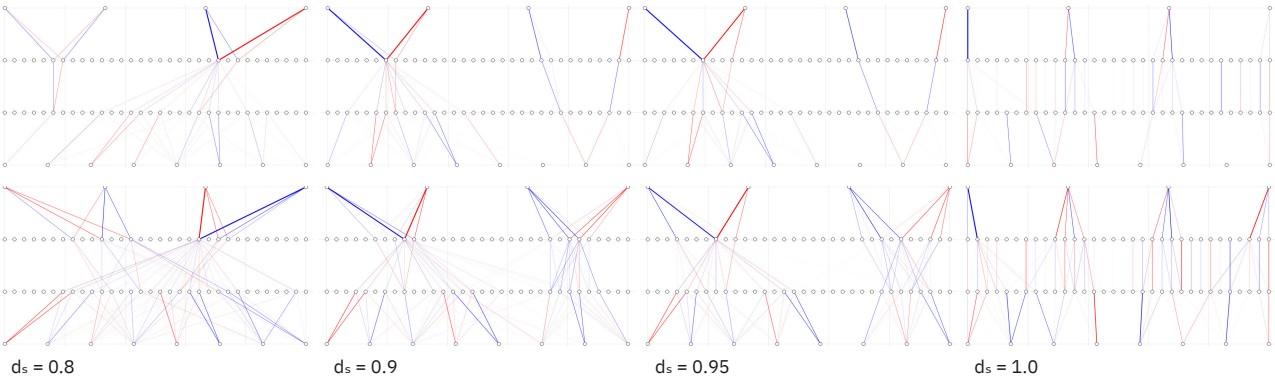

Figure 17: The impact of $d_s$ on the structure of the emergent modules (pre-pruning and fine-tuning). The networks shown have varying $\lambda_{cc}$ values, as the scale of the connection cost varies when $d_s$ is varied, but where selected to show networks with equivalent returns: top row $r \approx 0.75$, and bottom row $r \approx 0.82$.

We run all experiments in the main body of our work with $d_s = 0.95$, which was selected by comparing the relation between return, isolation and ARI of networks at different values. The results are plotted in Figure 18 and show that higher $d_s$ values result in higher isolation. Despite their high isolation score, we find that the modules detected in $d_s = 1$ networks align poorly with the correlation partitions: the resulting ARI values do not exceed 0.3 and do not increase monotonically with regularisation. We find that $d_s = 0.9$ offers the greatest ARI relative to return, but the difference between 0.8, 0.9 and 0.95 is relatively minor, so we select $d_s = 0.95$ for its significantly higher isolation scores.

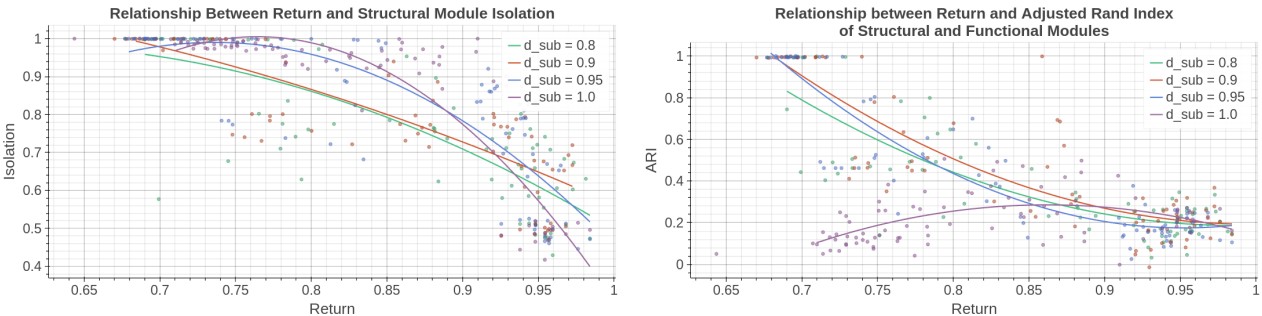

Figure 18: The impact of varying $d_s$ on the relationship between return and isolation (left) and between return and ARI (right) of the resulting networks' partitions.

## C.3. $\lambda_{cc}$ Scheduling

For the pruning fraction and $d_s$, we select the $\lambda_{cc}$ schedule based on the resulting relationships between return and isolation, and return and ARI. We select a linear introduction of the regularisation loss between 20% and 30% of training steps due its high relative performance with respect to these metrics, as shown in Figure 19.

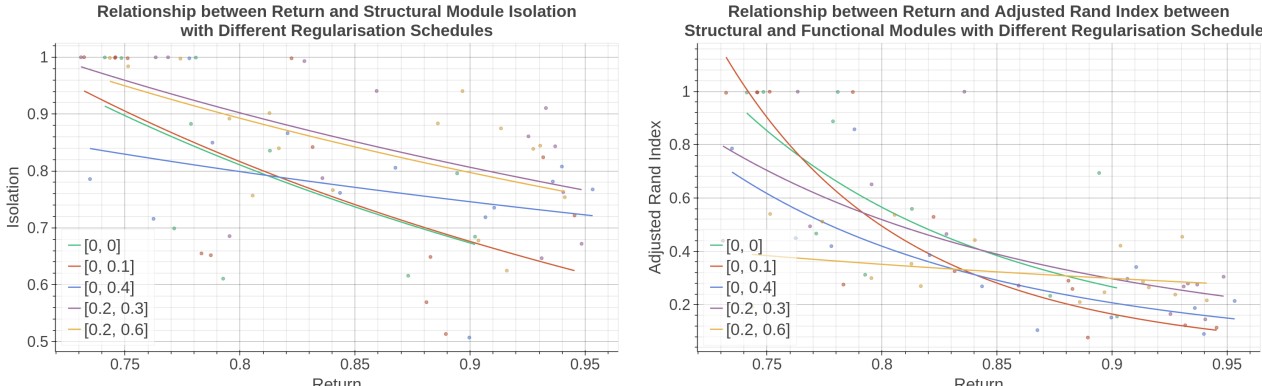

Figure 19: The impact of different introduction schedules for the CC loss on the relationship between return and isolation (left) and between return and ARI (right) of the resulting networks' partitions. A legend value of [x, y] indicates that $\lambda_{cc}$ was increased linearly between fractions x and y of the total training steps.

# D. Runtime Results

## D.1. Regularisation and Relocation

Figure 20 shows the average compilation and training times for 4M steps, comparing a vanilla PPO implementation with cases where distance weighted regularisation and neuron relocation are implemented in isolation and combined. We find an overall time increase in the combined case of 17%, which is dominated by the introduction of the connection cost loss.

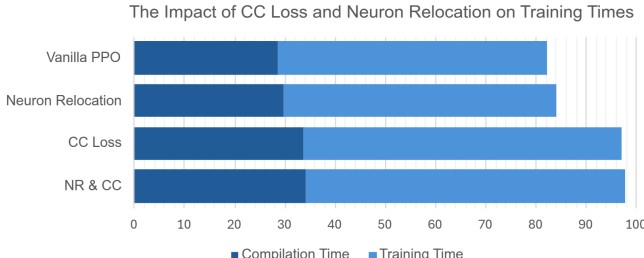

Figure 20: The average compilation and training times, for 4M training steps, of the Vanilla PPO implementation compared to PPO with neuron relocation, with distance weighted regularisation, and with both.

## D.2. The Extended Louvain Algorithm

The Louvain algorithm is commonly assumed to have a runtime complexity of $O(nlog(n))$[2][3](Huang et al., 2021) where $n$ is number of nodes. However, no definitive analysis of its time complexity has been performed, and other sources instead find a time complexity of $O(m)$ in the number of edges (Traag, 2015) . We compare the duration of the Louvain method to our proposed adaptations, with the caveat that this evaluation is limited by the relatively small scale of networks examined.

We observe that for our networks, the standard Louvain time complexity appears to match the $O(m)$ assumption. As our internal version simply applies Louvain but with fewer nodes and edges, it follows that it also has a linearly increasing duration with $m$, and this is what we observe in Figure 21. The figure further shows the duration of the fine-tuning stage, which also appears to be linear in $m$. A larger sample of networks, including larger ones, would be necessary to rigorously demonstrate this, however.

The calculation of the activation correlation matrix dominates the duration of our extended Louvain, and the duration of this is itself dominated by the model inference over the 10,000 episodes for which we collect activations, as shown in Figure 21. We note that we parallelise these episodes using JAX, so inference over 10,000 episodes takes negligibly longer than over a fewer number of episodes, but at 10,000 we reach the memory constraints of the 24GB GPU. Duration appears to increase slightly as the network size increases, but with multiple outliers. Should activation collections or correlation computations become prohibitive in terms of memory or computation as we scale to larger models, a number of approaches could be take to reduce the problem size. For example, we could take advantage of the localised nature of our modules, and separately collect and process activations for smaller network regions.

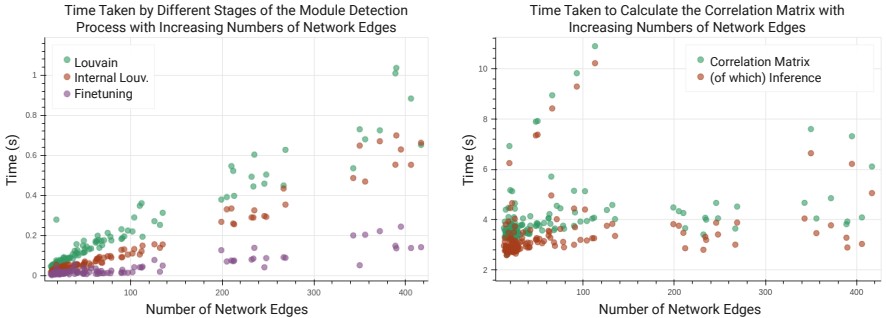

Figure 21: The observed duration of the Louvain algorithm and of the isolated components of our extended version.

---

[2]https://www.ultipa.com/document/ultipa-graph-analytics-algorithms/louvain/v4.5 (Accessed 9/01/25)
[3]https://perso.uclouvain.be/vincent.blondel/research/louvain.html (Accessed 9/01/25)

# E. Emergence of Modularity

## E.1. Additional Modularity Examples

We show two further examples of modularity emergence for each environment in Figure 22, and further show these networks partitioned using the fine-tuned internal Louvain in Figure 23. We find that the navigational modules emerge consistently and with the same structure in the DO and 3D-DO tasks. In the G2K task, we observe two structures: one which closely resembles the DO modules, and one where an action becomes disconnected from the network (but can still be selected based on its relative logit value compared to the remaining actions). As shown in Figure 23, we are still able to separate X and Y navigational modules controlling the remaining three connected actions.

We also show, in Figure 24, the impact of continuing to increase $\lambda_{cc}$ beyond the emergence of fully isolated modules: the prioritisation of goal information increases till these are the only features considered, but at a certain threshold we observe a collapse in both sparsity and return.

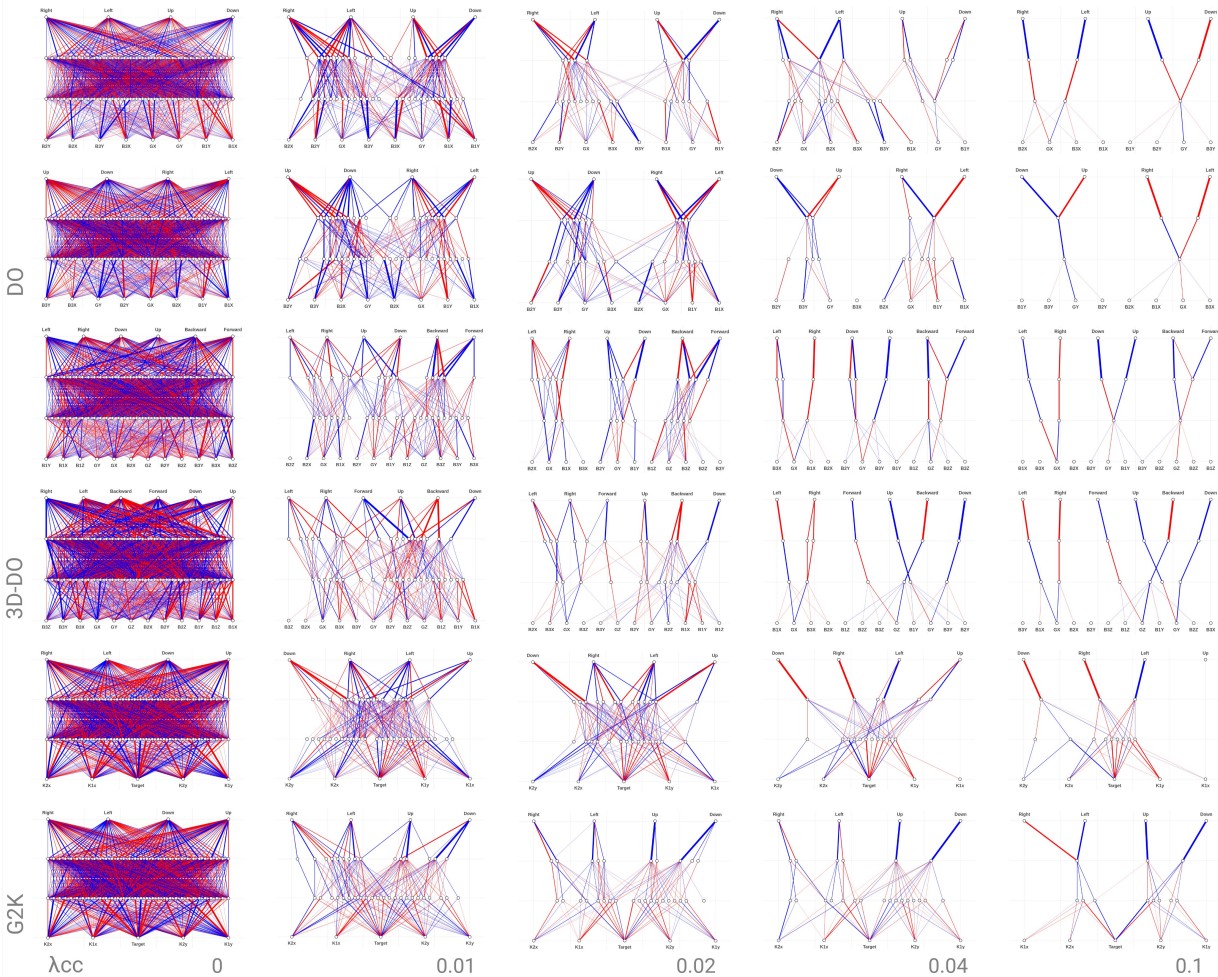

Figure 22: The emergence of modularity with increasing $\lambda_{cc}$ across two seeds in each environments.

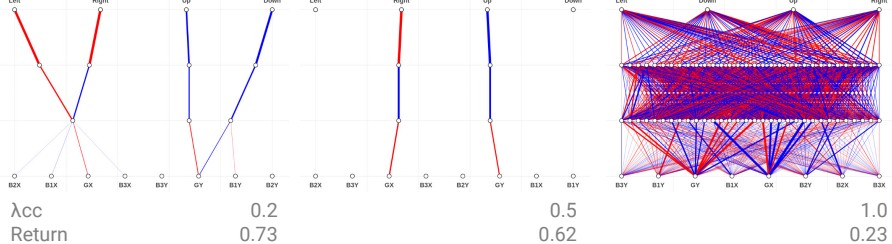

Figure 23: The networks shown in Figure 22 partitioned using our fine-tuned internal Louvain approach. The ordering of detected modules varies between $\lambda_{cc}$ values due to randomness in the order in which the Louvain algorithm considers nodes.

Figure 24: The network structures and returns observed when continuing to increase $\lambda_{cc}$ above the range considered in the main text.

### E.2. Isolating the Impacts of Regularisation, Distance and Relocation

While the emergence of visually distinct modules is evidently reliant on local connectivity induced by the connection cost loss and neuron relocation, we here analyse how these protocols contribute to the non-visual modularity measures. We conduct ablation experiments comparing networks trained with and without distance weighting and neuron relocation across different regularization strengths, and show how this affects module isolation and correlation alignment (ARI) in relation to return (Figure 25).

Naturally, networks without regularization ($\lambda_{cc} = 0$) have significantly less isolated modules than any regularized variant. We further find that both distance weighting and relocation increase isolation, but with a diminishing impact as regularisation increases. We expect this occurs occurs because strong sparsity constraints force isolated pathways regardless of their locality. The alignment (ARI) between weight-based and correlation-based partitions shows a different pattern: distance weighting but not relocation increases ARI scores, but in a manner that increases as regularisation increases. Examining sparsity, shown in Figure 26, we observe that distance weighting reduces sparsity relative to return, but relocation compensates for this, particularly at high regularisation levels. This suggests that relocation enables important but initially distant weights to be preserved, allowing sparsity to be achieved in a manner that is less damaging for return.

Overall, these results indicate that both distance weighting and relocation contribute to structural modularity, while distance weighting is particularly important for aligning structural and functional modularity. Although these effects are dependant on the strength of regularisation, both are observed in the most useful regularisation ranges where modularity is observed, but the impact on return is still relatively low. Intuitively, the distance weighting encourages additional sparsity and thus isolation by encouraging computations to be distributed across few weights, since each weight beyond the first weight is necessarily longer and more expensive. The neuron relocation likely enables to network to restructure in a manner that makes the weights with a greater performance impact shorter and thus less costly, promoting greater sparsity among less important weights (although weight is not a direct measure of importance it appears to be a relatively good proxy for it). This is particularly important given the connection cost scheduling: by the time sparsity is introduced, the network has already learnt effective computations which we wish to preserve. Further analysis will be required to fully understand and formalise how these impacts arise.

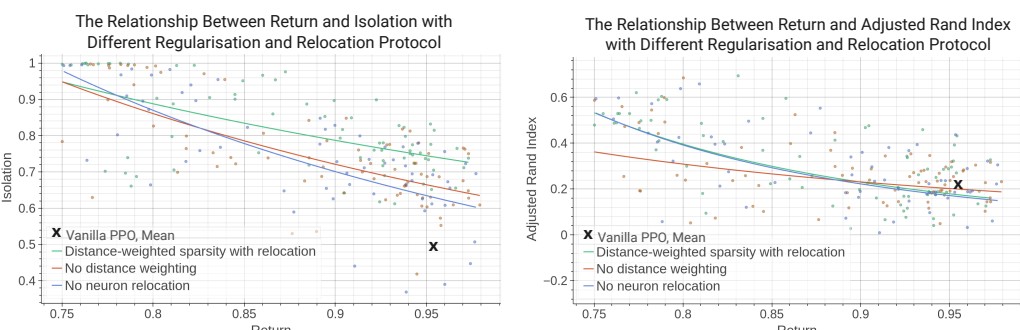

Figure 25: The relationships between return and isolation (left) and between return and ARI (right) with the distance weighting and neuron weighting separately ablated, across a range of $\lambda_{cc}$ values($[0.005, 0.1]$ where distance weighting is included and $[0.0005, 0.01]$ otherwise, to achieve equivalent regularisation levels and returns). We include the mean values achieved with the Vanilla PPO case, which corresponds to $\lambda_{cc} = 0$ and no relocation for comparison.

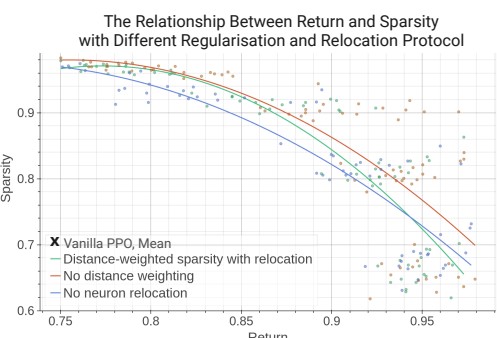

Figure 26: The relationships between return and sparsity, defined as the proportion of weights with a magnitude below 1% of the maximum magnitude in their layer, for the ablation cases considered in Figure 25. The vanilla PPO case has a significantly lower mean sparsity of 0.028, so is not shown.

# F. Pong Results

In this section we demonstrate the utility of our approach on Pong. We show that the sparse training protocol learns a single sparse module, which enables identification of a flaw in learnt Pong policies, as previously identified by Delfosse et al. (2024).

We train an MLP policy network on a custom implementation of Pong in JAX, which we modify to return symbolic observations. The observation consists of the agent paddle y position, the opponent paddle y position, the x and y positions of the ball and the x and y velocities of the ball. The opponent adopts the standard 'follow ball' policy and the agent receives sparse rewards of -1 and 1 when the opponent and agents score points respectively. We fix the regularisation parameters $d_s$, $k$ and the relocation intervals to use the same values as the Minigrid experiments, and conduct a small sweep over $\lambda_{cc}$ values. Full training parameters are shown in Table 2.

Table 2: PPO Hyperparameters

| ARCHITECTURE | |
| --- | --- |
| HIDDEN SIZE | 16 |
| NUMBER OF LAYERS | 2 |

| TRAINING | |
| --- | --- |
| PARALLEL ENVIRONMENTS | 16 |
| STEPS PER ENVIRONMENT | 128 |
| MINIBATCHES | 8 |
| EPOCHS | 16 |
| LEARNING RATE | 1E-5 |
| MAX GRADIENT NORM | 0.1 |
| GAE $\lambda$ | 0.99 |
| CLIP $\epsilon$ | 0.2 |
| ENTROPY COEFFICIENT | 0.01 |
| VALUE FUNCTION COEFFICIENT | 0.5 |
| TRAIN STEPS | 10 M |
| PRUNE FRACTION (APPENDIX C.1) | 0.01 |
| FINETUNE STEPS (APPENDIX C.1) | 10M |

| REGULARISATION | |
| --- | --- |
| $d_s$ (EQUATION 1) | 0.95 |
| K (SECTION 3.2) | 10 |
| RELOCATION INTERVAL (SECTION 3.2) | 2 |
| $\lambda_{cc}$ SCHEDULING (APPENDIX C.1) | 0.4-0.41 |

Unlike in the Minigrid tasks, the Pong agent moves along a single axis. We thus observe a single module in the computational graph, which becomes increasingly sparse as $\lambda_{cc}$ is increased, as shown in Figure 27. We show the impact of regularisation on agent performance and the number of network parameters in Figure 28. Up to a $\lambda_{cc}$ of 0.05, we observe a negligible impact on agent performance, with all policies with $\lambda_{cc} = 0.045$ achieving a perfect average score of 21. Beyond this we observe variability between seeds and a significant deterioration in performance in some cases. As in the Minigrid experiments, the regularisation and pruning also significantly reduces the number of parameters in the network, and we find we can achieve a mean game score of 21 with just 37 parameters.

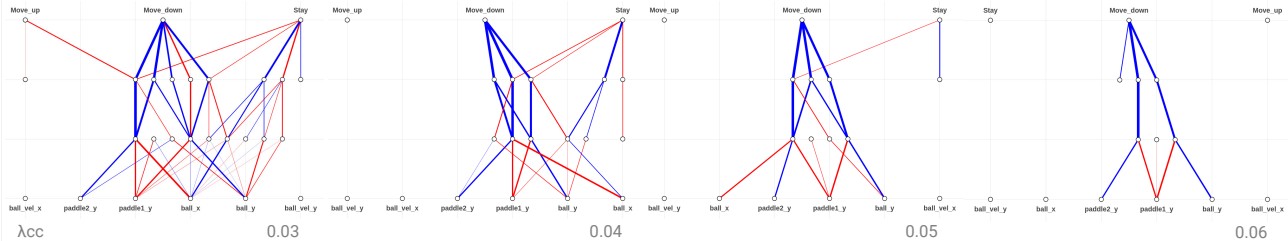

Figure 27: We observe sparsity increasing with $\lambda_{cc}$ in the Pong policy network. The simplicity of the task means a single module is learnt.

Delfosse et al. (2024) train Successive Concept Bottleneck Agents (SCBots) in the Pong environment and find a brittle reliance on the opponent position in the resulting policies. This is an artefact of the opponent policy, which attempts to keep

Figure 28: Left: The impact of $\lambda_{cc}$ regularisation on the mean player score at game end, with 21 indicating a 100% win rate. Right: The impact of $\lambda_{cc}$ regularisation on the number of network parameters post pruning, where the baseline non-regularised network uses 435 parameters.

the paddle centre aligned with the centre of the ball. While this results in high performance in training, it is an example of proxy gaming and is undesirable: if the opponent policy changes, the agent loses the ability to perform its target task of returning the ball.

Based on this observation, we retrain a set of policies with the opponent position removed from the observation space. The results, shown in Figure 29, show an increase in score variability at lower $\lambda_{cc}$ values, but still demonstrate the ability to achieve an 100% win rate up to a $\lambda_{cc} = 0.55$. By necessity, this policy relies solely on agent and ball information and is thus robust to the more realistic scenario of a variable opponent policy.

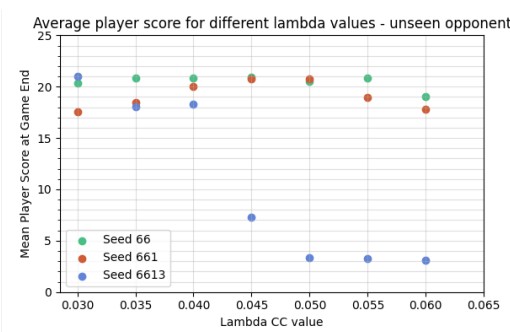

Figure 29: The impact of $\lambda_{cc}$ regularisation on the mean player score at game end, when the opponent position is removed from the observation space during training.

## G. Module Detection Method Data

Table 3: The Isolation and Functional Alignment (ARI) of actor network modules detected using the weights, correlation, internal, fine-tuned and fine-tuned internal versions of the Louvain algorithm. Results are averaged across 10 seeds in each of the DO, 3D-DO and G2K environment.

| | | ISOLATION | | | |
|---|---|---|---|---|---|
| $\lambda_{cc}$ | W Lv. | C Lv. | INT. | FT | FT INT. |
| 0.01 | 0.525 | 0.434 | 0.575 | 0.742 | **0.775** |
| 0.03 | 0.671 | 0.667 | 0.770 | 0.836 | **0.899** |
| 0.05 | 0.699 | 0.669 | 0.741 | 0.866 | **0.924** |
| 0.07 | 0.729 | 0.657 | 0.760 | 0.904 | **0.945** |
| 0.09 | 0.778 | 0.671 | 0.787 | 0.937 | **0.943** |
| 0.11 | 0.795 | 0.711 | 0.812 | 0.943 | **0.940** |
| 0.13 | 0.787 | 0.728 | 0.795 | **0.950** | 0.949 |
| 0.15 | 0.803 | 0.708 | 0.822 | **0.953** | 0.949 |

| | | FUNCTIONAL ALIGNMENT (ARI) | | | |
|---|---|---|---|---|---|
| $\lambda_{cc}$ | W Lv. | C Lv. | INT. | FT | FT INT. |
| 0.01 | 0.165 | - | 0.140 | **0.305** | 0.237 |
| 0.03 | 0.273 | - | 0.256 | **0.639** | 0.462 |
| 0.05 | 0.282 | - | 0.273 | **0.675** | 0.571 |
| 0.07 | 0.310 | - | 0.321 | **0.676** | 0.625 |
| 0.09 | 0.335 | - | 0.343 | 0.714 | **0.715** |
| 0.11 | 0.385 | - | 0.436 | 0.750 | **0.804** |
| 0.13 | 0.378 | - | 0.478 | 0.724 | **0.834** |
| 0.15 | 0.383 | - | 0.489 | 0.721 | **0.825** |

Table 4: The Isolation and Functional Alignment (ARI) of actor network modules detected using the weights, correlation, internal, fine-tuned and fine-tuned internal versions of the Louvain algorithm. Results are averaged across 10 seeds in the G2K environment only.

| | | ISOLATION | | | |
|---|---|---|---|---|---|
| $\lambda_{cc}$ | W Lv. | C Lv. | INT. | FT | FT INT. |
| 0.01 | 0.398 | 0.423 | 0.420 | 0.592 | **0.656** |
| 0.03 | 0.508 | 0.549 | 0.535 | **0.744** | 0.754 |
| 0.05 | 0.556 | 0.517 | 0.579 | **0.750** | 0.764 |
| 0.07 | 0.577 | 0.517 | 0.622 | 0.760 | **0.812** |
| 0.09 | 0.625 | 0.508 | 0.624 | 0.781 | **0.800** |
| 0.11 | 0.648 | 0.541 | 0.683 | **0.800** | 0.791 |
| 0.13 | 0.698 | 0.607 | 0.712 | **0.824** | 0.821 |
| 0.15 | 0.707 | 0.539 | 0.762 | **0.837** | 0.820 |

| | | FUNCTIONAL ALIGNMENT (ARI) | | | |
|---|---|---|---|---|---|
| $\lambda_{cc}$ | W Lv. | C Lv. | INT. | FT | FT INT. |
| 0.01 | 0.145 | - | 0.111 | **0.346** | 0.215 |
| 0.03 | 0.180 | - | 0.180 | **0.618** | 0.398 |
| 0.05 | 0.170 | - | 0.195 | **0.514** | 0.439 |
| 0.07 | 0.184 | - | 0.253 | **0.560** | 0.466 |
| 0.09 | 0.196 | - | 0.220 | 0.510 | **0.519** |
| 0.11 | 0.210 | - | 0.361 | 0.419 | **0.608** |
| 0.13 | 0.149 | - | 0.499 | 0.311 | **0.697** |
| 0.15 | 0.145 | - | 0.494 | 0.287 | **0.652** |

# H. Intervention Data

We present full action statistics for the networks interpreted in Section 4.6 showing the frequency of actions and their outcomes for the unmodified network, and for versions where modules are modified through negative saturation or negation. We bold the data corresponding to the axes the targeted community is associated with.

Table 5: Action Statistics for a 3D-DO network ($\lambda_{cc} = 0.06$, Figure 9a).

|  | Directions | Freq. | Failure | Success | Continue |
|---|---|---|---|---|---|
| Initial Network | up/down | 21.33% | 8.62% | 29.67% | 61.71% |
| Return = 0.77 | left/right | 42.55% | 8.29% | 21.70% | 70.01% |
|  | fwd/bwd | 36.12% | 9.81% | 37.48% | 52.71% |
| **Negative Saturation** |  |  |  |  |  |
| Community 0 | up/down | 40.74% | 2.11% | 1.20% | 96.69% |
| Return = 0.40 | left/right | 52.92% | 2.51% | 1.23% | 96.26% |
|  | **fwd/bwd** | **6.34%** | **3.12%** | **6.80%** | **90.07%** |
| Community 1 | up/down | 22.59% | 3.47% | 2.81% | 93.73% |
| Return = 0.45 | **left/right** | **6.18%** | **5.72%** | **14.34%** | **79.94%** |
|  | fwd/bwd | 71.23% | 3.49% | 2.03% | 94.48% |
| Community 2 | **up/down** | **6.02%** | **4.01%** | **15.16%** | **80.84%** |
| Return = 0.53 | left/right | 22.91% | 5.59% | 6.37% | 88.04% |
|  | fwd/bwd | 71.07% | 3.32% | 2.82% | 93.86% |
| **Negation** |  |  |  |  |  |
| Community 0 | up/down | 5.16% | 2.79% | 1.60% | 95.61% |
| Return = 0.14 | left/right | 11.14% | 2.43% | 1.12% | 96.45% |
|  | **fwd/bwd** | **83.70%** | **1.15%** | **0.02%** | **98.83%** |
| Community 1 | up/down | 0.99% | 10.03% | 4.47% | 85.50% |
| Return = 0.09 | **left/right** | **97.01%** | **1.11%** | **0.01%** | **98.88%** |
|  | fwd/bwd | 2.00% | 12.36% | 4.11% | 83.53% |
| Community 2 | **up/down** | **89.59%** | **1.00%** | **0.01%** | **99.00%** |
| Return = 0.39 | left/right | 3.53% | 8.48% | 13.27% | 78.26% |
|  | fwd/bwd | 6.87% | 5.92% | 8.15% | 85.93% |

Table 6: Action Statistics for a G2K network ($\lambda_{cc} = 0.12$, Figure 9b)

|  | Directions | Freq. | Failure | Success | Continue |
|---|---|---|---|---|---|
| Initial Network | up/down | 51.03% | 1.16% | 42.99% | 55.85% |
| Return = 0.94 | left/right | 48.97% | 3.57% | 28.96% | 67.47% |
| **Negative Saturation** |  |  |  |  |  |
| Community 0 | **up/down** | **2.38%** | **1.38%** | **12.57%** | **86.05%** |
| Return = 0.35 | left/right | 97.62% | 1.18% | 0.36% | 98.46% |
| Community 1 | up/down | 97.05% | 1.23% | 0.41% | 98.36% |
| Return = 0.39 | **left/right** | **2.95%** | **1.36%** | **13.64%** | **85.00%** |
| **Negation** |  |  |  |  |  |
| Community 0 | **up/down** | **98.64%** | **1.03%** | **0.01%** | **98.96%** |
| Return = 0.14 | left/right | 1.36% | 4.25% | 12.22% | 83.53% |
| Community 1 | up/down | 1.07% | 7.79% | 7.53% | 84.68% |
| Return = 0.08 | **left/right** | **98.93%** | **1.06%** | **0.01%** | **98.93%** |

Table 7: Action Statistics for a G2K network ($\lambda_{cc} = 0.02$, Figure 9c)

|  | Directions | Freq. | Failure | Success | Continue |
|---|---|---|---|---|---|
| Initial Network | up/down | 50.09% | 0.21% | 32.44% | 67.34% |
| Return = 0.99 | left/right | 49.91% | 0.41% | 40.51% | 59.07% |
| **Negative Saturation** |  |  |  |  |  |
| Community 0 | **up/down** | **4.78%** | **2.56%** | **7.06%** | **90.38%** |
| Return = 0.40 | left/right | 95.22% | 1.36% | 0.63% | 98.00% |
| Community 1 | up/down | 72.00% | 1.20% | 0.30% | 98.50% |
| Return = 0.16 | **left/right** | **28.00%** | **1.01%** | **0.01%** | **98.98%** |
| **Negation** |  |  |  |  |  |
| Community 0 | **up/down** | **78.39%** | **1.07%** | **0.57%** | **98.36%** |
| Return = 0.57 | left/right | 21.61% | 1.04% | 4.47% | 94.50% |
| Community 1 | up/down | 82.10% | 1.46% | 1.31% | 97.23% |
| Return = 0.49 | **left/right** | **17.90%** | **3.35%** | **3.63%** | **93.02%** |

