# OpenReview forum: "Inducing, Detecting and Characterising Neural Modules: A Pipeline for Functional Interpretability in Reinforcement Learning"
_ICML.cc/2025/Conference — ICML 2025 poster_

### Official Review · Reviewer_W7cH · 2025-03-07

**Overall Recommendation:** 3

**Summary:**

This paper addresses the challenge of interpretability in reinforcement learning (RL) models by proposing a method based on functional modules, aiming to overcome the scalability limitations of traditional neuron-level interpretability approaches. The authors introduce spatially aware regularization and neuron relocation techniques to promote weight sparsity and locality, thereby inducing the formation of functional modules within RL policy networks. Additionally, they extend the Louvain algorithm by incorporating a novel 'correlation alignment' metric to detect these modules effectively. Experiments conducted in 2D and 3D Minigrid environments demonstrate the emergence of distinct modules corresponding to different directional navigation tasks, with targeted interventions on network parameters validating the functional roles of these modules.​

**Claims And Evidence:**

The paper's primary claims are well-supported by empirical evidence. Notably, the authors demonstrate that increasing the strength of spatial regularization leads to more pronounced modular structures within the network. However, the functional validation of these modules relies on direct interventions in network weights, and the applicability of this approach in more complex or real-world scenarios remains to be fully established.

**Essential References Not Discussed:**

The paper's citations are comprehensive, covering essential works related to RL interpretability and neural modularity.

**Experimental Designs Or Analyses:**

The experimental design is robust, featuring ablation studies that elucidate the contributions of individual components, such as regularization and neuron relocation. However, the functional analysis methods—specifically, negative saturation and negation—are innovative but lack comparisons with existing interpretability techniques, potentially limiting the assessment of their effectiveness.​

**Methods And Evaluation Criteria:**

The proposed methods—including distance-based weight regularization, neuron relocation, the extended Louvain algorithm, and the correlation alignment metric—are clearly articulated and appropriate for the problem context. While the Minigrid environment serves as a suitable benchmark, its relative simplicity suggests that further validation in more complex tasks is necessary to assess the generalizability of the methods.​

**Other Comments Or Suggestions:**

1. While the figures are clear, enhancing the legends and descriptions could improve reader comprehension, particularly regarding the directional aspects of network structures.​

2. Conducting user studies to assess human understanding of the proposed modular interpretability approach could provide valuable insights into its practical utility.​

**Other Strengths And Weaknesses:**

Strengths:

1. Innovatively proposes a functionally interpretable framework for RL models based on neural modularity.​

2. Provides detailed technical implementations and rigorous experimental designs, enhancing reproducibility.​

3. Introduces novel metrics, such as correlation alignment, contributing valuable tools to the interpretability research community.​

Weaknesses:

1. The simplicity of the experimental environments may not fully capture the challenges present in more complex real-world tasks.​

2. The reliance on direct weight interventions for functional validation may limit applicability across diverse neural architectures.

**Questions For Authors:**

1. Applicability in Complex Scenarios: Can the negative saturation and parameter negation methods for module function validation be effectively applied to more complex, non-linear tasks? If so, what modifications would be necessary?​

2. Alternative Validation Methods: Have you explored activation-based approaches as alternatives to weight modification for module function validation? If not, could such methods offer advantages in certain contexts?​

3. Performance-Modularity Trade-off: Given the potential performance trade-offs associated with spatial regularization (λcc), have you investigated alternative strategies to mitigate performance loss while maintaining modularity?

**Relation To Broader Scientific Literature:**

The paper situates its contributions within the broader context of neural network interpretability research, particularly concerning modularity and hierarchical interpretability methods. It effectively references recent advancements in module detection and spatial regularization, highlighting its alignment with current research trends.

**Theoretical Claims:**

The paper does not present formal theoretical proofs; thus, there are no theoretical claims to evaluate in this context.

---

> ### Author Rebuttal · Authors · 2025-03-31
>
> Thank you for your detailed consideration of our work. We appreciate the acknowledgment of our robust experimentation and the insightful questions raised.
>
> **Figures and Captions.** We appreciate the advice on improving clarity. We have updated the network plots to include legends and have extended the captions to include sufficient detail for the core results to be understood from the figures and captions in isolation.
>
> **User Studies.** We agree that user studies can be valuable for assessing the utility of interpretability approaches. Although we agree with reviewer qymG that human evaluation is not necessary to support the claims in our work, future work leveraging our approach in specific applications would certainly be enhanced by context-specific user studies.
>
> **Function Validation: Complex Scenarios and Alternative Methods.** Our function validation approaches are not limited by the size or internal complexity of the modules. Our results instead show that the efficacy of ablation techniques in isolating module functionality are impacted by the level of connectivity between modules, which we regulate with the λcc parameter. We agree that activation-based intervention approaches may offer a valuable alternative, which we briefly discussion in the discussion section. Beyond this, we are excited about extending our approach to detect hierarchical modularity within complex tasks in future work.
>
> **Validation in Complex Scenarios.** We appreciate that Minigrid presents a relatively simple task set and that validation in more complex and real-world scenarios is a valuable direction for future work. To advance in this direction, we have extended our work to a non grid-world task, Pong, as proposed by reviewer qymG, and will incorporate the results into the camera ready paper.
>
> **Performance-Modularity Trade-Off.** We investigated a number of approaches to mitigating this, leading to the implementation of the λcc scheduling and the fine-tuning of the pruned modular networks (discussed in more detail in Appendix C), and the use of the log-based sparsity loss (discussed in Appendix A). We believe there may be further mitigation opportunities that merit testing in future work, for example by penalising inter-module connections and not intra-module connections in the later stages of training.

---

### Official Review · Reviewer_qymG · 2025-03-10

**Overall Recommendation:** 5

**Summary:**

The authors identify that most (post hoc) interpretability methods focus on explaining models' units (e.g. neurons), which does not scale. They propose to have interpretability at the level of *functional modularity*. They just try to identify neural modules, which are groups of neurons that are functionally related.
They use a "connection cost" lost to induce greater sparsity (than classic L1-loss) in the neural network, and perform neuron relocation.

They show that their method is able to induce interpretable modules  in a navigation task, and that these modules are more interpretable than the original network.

**Claims And Evidence:**

The main claims are clearly outlined in introduction.
The biggest claim is that local and sparse Neural Network are more interpretable than dense ones, which is commonly agreed upon in the litterature. No evidence (requiring human evaluation) is provided to support this claim, but I don't think that it is necessary. They also further explain: " High isolation implies minimal inter-module connectivity, resulting in stricter decomposability and enabling more independent module analysis.", which is a good argument for the modules' interpretability.

In introduction, some minor claims are not supported by citations:
"When considering its scaling to complex domains, RL interpretability must be considered at a level of abstraction which
balances tractability with fidelity to the underlying model." (l 43-45). Such claims could be removed.
In the introduction, a lot of litterature argue against the use of black-box neural networks, and favor the use of instrinsically interpretable models (e.g. decision trees, supported by Bastani). I would suggest to also spend the intro motivating the necessity of increasing the interpretability of neural networks (which is the main focus of the paper).

The results detailled in Functional Interpretability should somewhat be already highlighted in the captions of the figures. You can write Forward/Backward module in 9a.

**Essential References Not Discussed:**

While they are less essential, there are many published work on interpretable RL that should be discussed in RW.

**Experimental Designs Or Analyses:**

There are 2 major points of improvement here.
The first one is about clarity. The experimental section is quite hard to follow. A very detailled analysis has been done, but it is hard to follow. I would suggest to have a more structured approach to the experimental section.
Specifically, I would suggest to have list of scientific questions at the beginning of the experimental evaluation section (often denoted Q1, Q2, ... etc), that are each answered in different paragraphs. For example
Q1/ Can more interpretable modules reach similar performances as non constrained baselines?
Q2/ Does our algorithm induce more interpretable modules than the original network? (maybe name your agents that include every interpretability inducing techniques)
Q3/ How does the regularisation parameter affect the performance and interpretability of the modules?
Q4/ other ablation studies.

The second is to also evaluate their method on a non maze environment.
The paper does not limit the methods' interpretability to mazes, maybe the evaluation should go beyond navigation tasks. (Maybe MinAtari (JAX implemented) or OCAtari ?).
I think that training one network on the OCAtari version of Pong might to similarly interpretable networks.
The input space is 6 dimensional (x and y coordinates for each relevant objects).
You could identify if e.g. the constant x positions of the Player and the enemy only have nul weights (as they are constant and thus do not lead to any information gain). You might be even able to get an insight to the Pong misalignment problem with your technique (by analyzing that the module focusing on the enemy's vertical position is having high weights).

Now, I know how work demanding it is to include yet another evaluation to the paper, but I think that it would be a great addition to the paper.

I was hesitating between weak reject and weak accept, but if all my concerns are adressed (or for the additional experiments, that could take time, if the authors start them and will include it in the camera ready version), I am willing to increase my score to strong accept if my concerns (also the ones bellow) are addressed in the rebuttal, as I think that the paper is very strong and has a lot of potential.

A more minor point: What do the communities stand for in e.g. Figures 5 and 6? What is the meaning of the colors? I would suggest to add a legend to the figures. Can you interpret the communities? (e.g. the red community is responsible for the vertical agent's movement)? This should go into the captions of the figures.

For the experimental results, I would suggest structuring the figures like this:
The first sentence should highlight the main message of the Figure/Table (e.g. "*Fine-tuned Internal Louvain* allow for more isolated (thus interpretable) modules than the other approaches.").
The next sentences then explain what is depicted in the Table/Figure. E.g. "Communities are detected by the Louvain algorithm, and the colors represent the different communities. The *Internal* variant leads to denser communities, *Fine-tuned* leads to less communities, which is a sign of more interpretable modules. The top row depicts ... while the bottom row depicts ... ."
Finally, details and references to e.g. appendix can be provided if necessary. E.g. Results on more environmnets are provided in appendix X.

This would greatly improve the readability of the paper. I personally tend to read the abstract and the figures first, and then the rest of the paper. If the figures are well structured, I can get a good understanding of the paper without reading the whole paper (and thus decide if I potentially dive into it). I know that this is also the case for many other readers.

**Methods And Evaluation Criteria:**

The proposed method:
* brings local and sparse networks to RL.
* proposes to use Logarithmic Sparsity Loss to increase even further sparsity in the network.
* extends the Louvain algorithm to detect modules in the network.


The authors evaluate their method on a navigation task, but only report interpretability, no performance metric is reported. Before checking for the level of interpretability of agents, we first need to check to which extent they learned to solve the task. I have not been able to find completion rate, or any other any performance metric in the paper.
In *Limitations and Future Work*, the authors mention that "*λcc controls this balance*", but I do not see any results on how the performance of the agents is affected by the value of λcc.

**Other Comments Or Suggestions:**

The paper could benefit from a more detailed discussion on the benefit and the intuition behind the Neuron Position Optimization.

I would bring algorithm 2 before the start of the experimental section and e.g. adjust the vertical space around the equations to have section 4 start on page 4. (This is very final formatting, but it would make this great paper even more appreciable). But the paper might heavily change, so this might not be accurate.

**Other Strengths And Weaknesses:**

I mostly checked appendix A, which gives a nice intuition on the log-based sparsity loss, and appendix D for the runtime results, but globally looked up everything.

**Questions For Authors:**

* You apply Neuron Position Optimization every T steps. Why do you need to apply it every T steps? Why not only at the end?
* How does the empirical computational cost grow with the size of the problem? Is it feasible to apply it to larger scale problems?
* Are you the first to propose a logarithmic sparsity loss? (This is quite an simple amazing idea, but I am not sure if it is novel).
* Can you provide more details on why the log-based sparsity loss is preferable to the exponential-log one? I have read appendix A, but
* Can you add scores in Figure 2 for the different methods on each task? Maybe next to $\lambda_{cc}$ ?
* What part of your algorithm is transferable to CNNs? (Neuron relocation? Louvain Algorithm?)
* Are there some modules that are not easily interpretable? Would one need to apply a post-hoc interpretability method to understand them? This would not be a reason to reject, but would be great to understand the limits of your method.


Intro could be shortened, directly go to the point (particularly if you create a *Related Work* section).
l 39: "to improving" -> "to improve"
l 10: "which directly implicate on" -> "which directly impact"

**Relation To Broader Scientific Literature:**

This is the 3rd major opportunity for improvement.
As said, I would rewrite the experimental section to have a more structured approach to the evaluation.
Thus, you could add a related work section that discuss interpretability and sparsity in RL (could be merged with the dicussion). I tend to place background before the method (as the background is necessary to understand the method), and the related work (i.e. other approaches to the same problems: interpretability and sparsity) after the evaluation of your method, to avoid having the readers biased towards thinking of other potential approaches while presenting yours.

You are missing many of the latest published Related Work on Interpretable RL. I hereafter provide a list of papers that should be included in your related work section:

* Delfosse et al. "Interpretable concept bottlenecks to align reinforcement learning agents." NeurIPS (2024).

* Luo et al. "End-to-End Neuro-Symbolic Visual Reinforcement Learning with Language Explanations." ICML (2024).

* Kohler et al. "Interpretable and Editable Programmatic Tree Policies for Reinforcement Learning." RLC workshop (2024).

* Delfosse et al. "Interpretable and explainable logical policies via neurally guided symbolic abstraction." NeurIPS (2024).

* Marton, et al. "SYMPOL: Symbolic Tree-Based On-Policy Reinforcement Learning." ICLR (2025).

* Shindo, et al. "BlendRL: A Framework for Merging Symbolic and Neural Policy Learning." ICLR (2025).

**Theoretical Claims:**

Not Applicable.

---

> ### Author Rebuttal · Authors · 2025-03-31
>
> Thank you for your detailed and thoughtful review. We are grateful for your recognition of the potential of our work and the constructive feedback which we have applied as follows. We have also updated the figures and draft PDF on the project page to reflect the changes: https://sites.google.com/view/mod-xrl/home
>
> **Additional Experiments.** We agree that a non-maze application would strengthen our results and appreciate the Pong suggestion - thank you! We have adapted the Gymnax version to return a symbolic observation, and have implemented the distance-weighted sparsity training. Initial results show this learns a single sparse module that uses only a subset of the inputs and actions. The agent focuses on the opponent even at high sparsity, and we are excited to conduct ablations to see if this offers insights into the Pong misalignment problem. We have added these early results to the project page, and will include complete findings in the camera-ready paper.
>
> **Structure and Clarity.** We appreciate these helpful suggestions. We agree the claim regarding levels of abstraction is unecessary and have instead expanded the argument for the utility of interpretability. We have added task scores to Figure 2 and legends to Figure 9 and have extended all captions such that the results shown are now broadly understandable in isolation from the text. We have added a set of research questions to guide the Experiments section, reduced the Background to contain only necessary information and extended the removed information into a Related Work section. Thank you for the related papers, particularly Delfosse et al. (2024) and Kohler et al. (2024), which provide valuable examples for the utility of RL interpretability. We have included these in the Introduction and the other papers in Related Works. We have retained half a page for the Pong results, but may have to move a portion of these or the Related Works to the Appendices to meet the 9 page limit.
>
> **Performance Metric.** Since the reward is 1 on task completion and 0 otherwise, the return in Figure 8 is equivalent to both the success rate and average reward. We have clarified this in the text and caption.
>
> **Neuron Position Optimisation Intuition.** We appreciate your interest and describe our intuition below. We have added this to the Methods and App. E.2. We consider the distance weighting as encouraging computation to distribute across few weights and neurons, as each additional weight used is 'more expensive' than using the single shortest weight. Since we schedule λcc, sparsity is introduced when the network already implements relevant computations. Position optimisation thus allows existing important weights to move and become 'less expensive', while less important ones are more heavily penalised by the CC loss. We expect this same intuition holds without scheduling, as initial weights will bias learning towards specific local optima in the parameter space. Regarding the **frequency of position optimisation**: applying it every T steps improves module isolation and ARI (as reported in App. E.2). These metrics are artefacts of the learnt network connectivity, so we would not obtain the same benefits from relocating neurons after training.
>
> **Computational Cost.** The complexity of the connection cost calculation is linear with depth and quadratic with width. Relocation is linear with layers and cubic with width, but this can be mitigated by reducing the number of swaps considered or increasing T. We trained a set of networks with increasing widths (32 to 512) and depths (2 to 50). With increasing width, the time increase due to the CC loss remains a relatively constant percentage of train time (15-19%), whereas the relocation percentage increases from 1% to 7% , which supports the theoretical complexity. We present network partitioning complexity results in App. D2.
>
> **Log-loss Novelty.** We are unaware of prior work using a logarithmic sparsity loss, but can't definitively state that it has not been applied in contexts we have not identified.
>
> **Exponential Log-loss.** We initially ran a small set of experiments with the exponential formulation and did not observe any differences in results. Since we trained networks with different numbers of weights, in hyper-parameter tuning and due to varying input and output dimensions, the non-exponential formulation was more desirable, as it is additive with number of weights and made selecting appropriate λcc ranges straightforward.
>
> **CNN Application.** For conciseness, we point you to our response to reviewer jddf.
>
> **Module Interpretability.** We did not come across any non-interpretable modules, but agree this is an interesting point and that high level modules in complex networks may not be immediately interpretable using ablations. We are excited about the potential to address this by applying our modularity technique in a hierarchical manner, which may enable the extraction of interpretable submodules.

---

> > ### Comment · Reviewer_qymG · 2025-04-03
> >
> > On the additional experiments, the Pong misalignment issue has been detected on the (OC)Atari version of the Pong ALE environment, I am not sure if it is also present in gymnax. Do you know if the p1.x and p2.x are also constant in the gymnax version? It seems that the sparse model do not attach importance to these.
> >
> > Thank you for your clarifications, I think that the discussion on the Neuron Position Optimisation Intuition and on the cost should be included in the paper if possible (at least in appendix and be referred to).
> >
> > Apart from this, I find your answer and the overall approach quite inspiring, as it constitute a good first step to the development of more interpretable neural components for RL agents.
> >
> > I am raising my score. Many thanks for this inspiring work!

---

> > > ### Author Response · Authors · 2025-04-04
> > >
> > > Thank you for your appreciation of our work!
> > >
> > > We do observe the same phenomenon as in OCAtari where the network attaches importance to the opponents y position. We will investigate performance in the No Enemy and Lazy Enemy cases proposed by Delfosse et al. (2024) and discuss this in the final paper.
> > >
> > > P1x and P2x are constant in Gymnax as in (OC)Atari, but interestingly we find that the sparse network attaches a weak importance to the opponent's x position. This may reduce with continued training.
> > >
> > > We will also include the Neuron Position Optimisation intuition briefly in the Methods, referencing the full description in the Appendix. Many thanks again for your helpful suggestions to enhance and clarify our work.

---

### Official Review · Reviewer_jddf · 2025-03-13

**Overall Recommendation:** 3

**Summary:**

This paper presents a pipeline for inducing, detecting, and characterizing neural modules within reinforcement learning (RL) policy networks to enhance interpretability. By penalizing non-local connectivity and encouraging sparsity and locality in network weights, the fully connected networks used in the study exhibit functional modules. To automatically detect these modules, they extend the classical Louvain community detection algorithm by incorporating a “correlation alignment” metric that accounts for the unique architectural constraints of neural networks. The method is validated in both 2D and 3D Minigrid environments, where distinct navigational modules emerge that correspond to specific movement axes. Furthermore, the paper demonstrates that targeted interventions—such as disabling or perturbing specific modules—can empirically confirm their functional roles. Overall, the work offers a framework for decomposing and understanding complex RL decision-making processes through functional modularity.

**Claims And Evidence:**

The paper’s claims are generally well-supported. It convincingly shows that spatial regularization and neuron relocation lead to the emergence of functionally coherent modules and that its extended Louvain method reliably detects these modules. Interventions validate that these modules serve distinct functions, and the trade-off between improved interpretability (and sparsity) and a modest performance drop is clearly demonstrated. However, while the results in simple 2D and 3D Minigrid tasks are promising, the **scalability claim** is supported only in these limited settings on fully connected networks, suggesting that further evidence on more complex tasks is needed to fully substantiate scalability.

**Essential References Not Discussed:**

Non detected

**Experimental Designs Or Analyses:**

Not very toughly, I am not much familiar with Minigrid tasks or the Louvain algorithm.

**Methods And Evaluation Criteria:**

Methods and evaluation criteria is the main strength of this paper, from inducing, to evaluation, and finally to knock out studies, all seem fine to me.

**Other Comments Or Suggestions:**

None

**Other Strengths And Weaknesses:**

The paper’s approach of imposing spatial correlations is conceptually compelling and aligns well with neuroscience theories explaining feature maps in the visual cortex (as cited). This integration of spatial regularization into reinforcement learning offers enhanced interpretability and a clear mechanism for module emergence. However, a notable drawback is that this spatial constraint appears to cap model performance, as evidenced by previous studies and the modest accuracy drop reported here. Consequently, while the model yields solutions that are more interpretable, it may preclude discovering the high-performing strategies that less constrained, more powerful networks can achieve. This trade-off highlights a broader tension in mechanistic interpretability research—balancing the need for clear, interpretable mechanisms against the pursuit of state-of-the-art performance on complex tasks.

**Questions For Authors:**

To what extent the pipeline depends on the MLP architecture of the models tested? Can one apply the same pipeline for CNN or transformers?

**Relation To Broader Scientific Literature:**

The paper tries to bridge RL and interpretability literatures which make it well suited for broader scientific literature.

**Theoretical Claims:**

N/A

---

> ### Author Rebuttal · Authors · 2025-03-31
>
> Thank you for your detailed response and consideration of our work. We appreciate your recognition of the compelling nature of our proposed approach, and respond to the points raised as follows:
>
> **Scalability.** We agree that application to more complex tasks will be valuable for further evidencing scalability. Our automated module detection techniques, in particular, provide a robust foundation for this future work. As proposed by reviewer qymG, we have now implemented the framework on a non grid-world environment (Pong) and will formalise and incorporate these results in the final paper.
>
> **Interpretability vs Performance Trade-off.** We acknowledge this trade-off, which, as you note, is a tension observed both in our work and the broader interpretability field. One advantage of our approach is the ability to moderate this balance using the regularisation factor λcc. This differs from other white-box approaches like decision trees and offers a valuable means of tailoring the interpretability-performance trade-off for different use cases.
>
> **Extension to Alternative Architectures.** Thank you for raising the potential for extension to CNNs or transformers. We agree this is a valuable future research direction and share some preliminary thoughts about extending our work in this direction:
>
> Considering CNNs, the sparsity and distance metrics are not obviously applicable to standard kernel computations, but there is potential to consider distance and sparsity metrics in branched architectures (such as the Inception models). Conceptually, the notion of functional modularity seems more applicable to decision-making than image-processing tasks. It may thus be more interesting to interpret modules within fully connected layers downstream of convolutional layers, and this may also improve the interpretability of the intermediate latent space.
>
> Considering transformers, our modularity pipeline could be applied (with adaptation) by taking attention heads as network nodes or, as with CNNs, by interpreting MLP layers only. Alternatively, we could frame parameters as nodes and relocate vectors within attention head matrices and positions within the residual stream.

---

### Official Review · Reviewer_s33i · 2025-03-14

**Overall Recommendation:** 3

**Summary:**

This paper proposes a method to learn a functionally modular and interpretable model in an RL policy network. It combines a few ideas:

- Spatially embedded neurons with a distance-weighted loss to encourage locality
- Neuron relocalization (Algorithm 1)
- Partitioning the model into different modules using variants of the Louvain algorithm that take into account functional connectivity (equation 5)

The paper tests these ideas on a few GridWorld RL environments, and find that the discovered modules are interpretable; when the modules are intervened upon, they display effects which are consistent with their modular nature.

**Claims And Evidence:**

The paper claims to:

- extend local neural networks methods to RL
- create an extended Louvain algorithm to detect communities in the neural networks
- demonstrate that interventions on the network parameters that are derived from the Louvain algorithm are consistent with their purported roles

These are fairly thin claims–it seems to me like a straightforward application of prior methods on a slightly different problem, and strictly speaking their extensions are not necessary to make the method work specifically in the RL setting, they're more like enhancements of the methods in general–but they are well supported.

**Essential References Not Discussed:**

N/A

**Experimental Designs Or Analyses:**

The experiments are sound and appropriate.

**Methods And Evaluation Criteria:**

The methods and evaluation criteria are appropriate. These are still very much toy problems to demonstrate the method. If the ultimate goal is to apply this to non-trivial tasks, as the introductory framing in terms of AI ethics implies, one would want to extend this to larger and more complex environments.

As is often the case in interpretability research, the claims of finding insights are highly subjective. Their interventions in 4.5 partially address this issue, but it doesn't convince me that this will scale to non-trivial problems.

**Other Comments Or Suggestions:**

There's a few typos:

- "Comission"
- "cna"
- "combine" rather than combined on line 399

There's a sentence that doesn't make sense: "negative saturation of modules evidences their axes specific navigation function"

I don;t like the claim that conscious decision making relies on modular processing–it seems unnecessary to drag consciousness into this.

**Other Strengths And Weaknesses:**

This paper feels like a rather straightforward extension of prior work. It is well executed, however. My preference is to evaluate the content on execution rather than the more subjective issue of originality.

I found the Background and Related works section to be particularly well-executed and far-reaching.

**Questions For Authors:**

N/A

**Relation To Broader Scientific Literature:**

This is a straightforward extension of Liu et al. (2023) to RL with a few bells and whistles to find good networks via the Louvain algorithm.

**Theoretical Claims:**

N/A

---

> ### Author Rebuttal · Authors · 2025-03-31
>
> Thank you for the thorough review. We appreciate the positive comments on the quality of our execution and are grateful for the constructive feedback, which we respond to below.
>
> **Contributions.** We recognise that our training approach builds on Liu et al (2023). Specifically, we do this by adapting distance-weighted sparsity to RL policies rather than tasks with explicit mathematical structure, and by proposing extensions including log-based sparsity to improve performance. We further develop a novel clustering approach and propose modularity metrics which enable automated detection and characterisation of functional neural modules. We believe this is a crucial step to enabling scalable module interpretability.
>
> **Interpretability Insights and Scalability.** We appreciate your understanding of the difficulties of finding objective insights in interpretability research, a challenge which we approached by quantifying the impact of modules through ablations. We agree that demonstration in more complex environments is an important direction for future work. To move towards this, we have now implemented the framework on a non grid-world environment (Pong), as proposed by reviewer qymG, and will formalise and incorporate these results in the final paper. We believe that our rigorous evaluation approach, particularly the ablation studies and hyperparameter analyses, will provide a solid methodological foundation for further scalability.
>
> **Specific Corrections.** Thank you for drawing the typos to our attention - we have fixed these. The sentence about negative saturation was intended to explain that when modules are negatively saturated, the modified policy behaviour shows evidence of the modules' axis-specific navigation functions. We have clarified this sentence. We also acknowledge that the claim about consciousness is unnecessary to motivate our work and have removed it.

---

### Decision · Program_Chairs · 2025-05-01

**Decision:**

Accept (poster)

**Comment:**

This paper proposes a mechanism to produce interpretable RL policies via network locality and sparsity, and an extension of a pre-existing Louvian algorithm. Although there are some concerns regarding the limited type of environments and incremental nature of the work, all reviewers support acceptance, and I am in agreement with them.

I would urge the authors to make sure they address the reviewer feedback for the final version, in particular with regards to the extra Pong experiments, as well as the clarity of the empirical presentation.

Nit: The title should probably be "Inducing, Detecting and Characterising Neural Modules: A Pipeline for Functional Interpretability in Reinforcement Learning" (changing "Functionally" to "Functional").